# Localization of KRAS downstream target ARL4C to invasive pseudopods accelerates pancreatic cancer cell invasion

Akikazu Harada[1,2], Shinji Matsumoto[1,2], Yoshiaki Yasumizu[2,3], Kensaku Shojima[1,4], Toshiyuki Akama[1], Hidetoshi Eguchi[5], Akira Kikuchi[1,2]*

[1]Department of Molecular Biology and Biochemistry, Graduate School of Medicine, Osaka University, Suita, Japan; [2]Integrated Frontier Research for Medical Science Division, Institute for Open and Transdisciplinary Research Initiatives (OTRI), Osaka University, 2-2 Yamadaoka, Suita, Japan; [3]Laboratory of Experimental Immunology, WPI Frontier Immunology Research Center, Osaka University, Suita, Japan; [4]Gene Expression Laboratory (GEL-B), Salk Institute for Biological Studies, San Diego, United States; [5]Gastroenterological Surgery, Graduate School of Medicine, Osaka University, Suita, Japan

*For correspondence:
akikuchi@molbiobc.med.osaka-u.
ac.jp

**Abstract** Pancreatic cancer has a high mortality rate due to metastasis. Whereas KRAS is mutated in most pancreatic cancer patients, controlling KRAS or its downstream effectors has not been succeeded clinically. ARL4C is a small G protein whose expression is induced by the Wnt and EGF–RAS pathways. In the present study, we found that ARL4C is frequently overexpressed in pancreatic cancer patients and showed that its localization to invasive pseudopods is required for cancer cell invasion. IQGAP1 was identified as a novel interacting protein for ARL4C. ARL4C recruited IQGAP1 and its downstream effector, MMP14, to invasive pseudopods. Specific localization of ARL4C, IQGAP1, and MMP14 was the active site of invasion, which induced degradation of the extracellular matrix. Moreover, subcutaneously injected antisense oligonucleotide against ARL4C into tumor-bearing mice suppressed metastasis of pancreatic cancer. These results suggest that ARL4C–IQGAP1–MMP14 signaling is activated at invasive pseudopods of pancreatic cancer cells.

## Introduction

Pancreatic cancer is extremely aggressive and exhibits poor prognosis, with a 5 year survival of only 5 % (*Klein, 2013*). Most pancreatic cancer-related deaths are due to metastatic disease, and more than 80 % of patients have either locally advanced or metastatic disease (*Hidalgo, 2010*; *Klein, 2013*). Genome sequencing analysis has revealed the mutational landscape of pancreatic cancer and KRAS mutations are considered an initiating event in pancreatic ductal cells (*Collins et al., 2012*; *Waddell et al., 2015*). Irrespective of our improved understanding of tumor biology, the treatment outcome has not changed for many years. Therefore, new innovative treatment options need to be tested based on better understanding of the characteristics of pancreatic cancer.

ARL4C is a member of the ADP-ribosylation factor (ARF)-like protein (ARL) family, which belongs to the ARF protein subgroup of the small GTP-binding protein superfamily (*Engel et al., 2004*; *Matsumoto et al., 2017*; *Wei et al., 2009*). Cytohesin2/ARF nucleotide-binding site opener (ARNO), a GDP/GTP exchange factor of ARF family proteins, has been identified as a direct effector protein (*Hofmann et al., 2007*). ARL4C is expressed through activation of Wnt–β-catenin and EGF–RAS signaling and plays important roles in both epithelial morphogenesis and tumorigenesis (*Matsumoto et al., 2017*; *Matsumoto et al., 2014*). Because aberrant activation of the Wnt–β-catenin and/or EGF–RAS

**eLife digest** Most cases of pancreatic cancer are detected in the later stages when they are difficult to treat and, as a result, survival is low. Over 90% of pancreatic cancers contain genetic changes that increase the activity of a protein called KRAS. This hyperactive KRAS drives cancer growth and progression. Attempts to treat pancreatic cancer using drugs that reduce the activity of KRAS have so far failed.

The KRAS protein can accelerate growth in healthy cells as well as in cancer and it does this by activating various other proteins. Drugs that target some of these other proteins could be more effective at treating pancreatic cancer than the drugs that target KRAS. One of these potential targets is called ARL4C. ARL4C is active during fetal development, but it is often not present in adult tissues. Harada et al. investigated whether the protein is important in pancreatic cancer, and what other roles it has in the body, to better understand if it is a good target for cancer treatment.

First, Harada et al. used cells grown in the lab to show that ARL4C contributes to the aggressive spread of human pancreatic cancers. Using mice, Harada et al. also showed that blocking the activity of ARL4C in pancreatic cancers helped to slow their progression.

Harada et al.'s results suggest that ARL4C could be a good target for new drugs treating pancreatic cancers. Given that this protein does not seem to have important roles in the cells of adults, targeting it is unlikely to have major side effects. Further investigation of ARL4C in more human-like animal models will help to confirm these results.

pathways are frequently observed in various types of cancers, ARL4C is indeed expressed in a number of cancers (*Fujii et al., 2015*; *Fujii et al., 2016*). In colon and lung cancer cells, ARL4C promotes cell proliferation through ARF6, RAC, RHO, and YAP/TAZ. On the other hand, in liver cancer cells, ARL4C promotes cell proliferation through phosphatidylinositol three kinase δ (PI3Kδ) (*Harada et al., 2019*). Thus, ARL4C would activate different downstream pathways in a cancer cell context-dependent manner. These prompted us to study the involvement of ARL4C, as a KRAS downstream molecule, in aggressiveness of pancreatic cancer, and IQ-domain GTPase-activation protein 1 (IQGAP1) was identified as a binding protein of ARL4C.

IQGAPs are an evolutionarily conserved family of proteins that bind to a diverse array of signaling and structural proteins (*Hedman et al., 2015*). Mammalian IQGAP1 is a well-characterized member of the IQGAP family and a fundamental regulator of cytoskeletal function (*Briggs and Sacks, 2003*). IQGAP1 is highly expressed in the tumor lesions and suggested to be involved in cancer cell metastasis (*Johnson et al., 2009*; *Sakurai-Yageta et al., 2008*). Here, we show that ARL4C bound to IQGAP1 and recruited IQGAP1 and membrane type1-matrix metalloproteinase (MT1-MMP, also called MMP14) (*Sakurai-Yageta et al., 2008*) to invasive pseudopods in a phosphatidylinositol (3,4,5)-trisphosphate (PIP3)-dependent manner and accelerated invasion. In addition, ARL4C antisense oligonucleotide (ASO) suppressed the lymph node metastases of pancreatic cancer cells orthotopically implanted into the pancreas of immunodeficient mice. These results suggest that the ARL4C–IQGAP1–MMP14 signaling axis promotes pancreatic cancer aggressiveness and that ARL4C is a novel molecular target for the treatment of pancreatic cancer.

## Results

### ARL4C is expressed in human pancreatic cancer

Whether ARL4C is expressed in pancreatic cancer patients was examined using immunohistochemistry. Fifty-seven pancreatic ductal adenocarcinoma (PDAC) patients, who did not receive preoperative chemotherapy, were used in this study (*Supplementary file 1 table 1*; *Source data 1*). ARL4C staining in the tumor lesions was calculated as a continuous variable, and the patients were classified into two groups (high and low), depending on ARL4C expression levels (*Figure 1A*). ARL4C expression was considered high when the total area of the tumor stained with anti-ARL4C antibody exceeded 5 %. High expression of ARL4C was observed in 47 cases (82%), but minimally detected in non-tumor regions of pancreatic ducts (*Figure 1A*). Anti-ARL4C antibody used in this study was validated in western blotting and immunohistochemical assay (IHC) (*Figure 1—figure supplement 1A and B*).

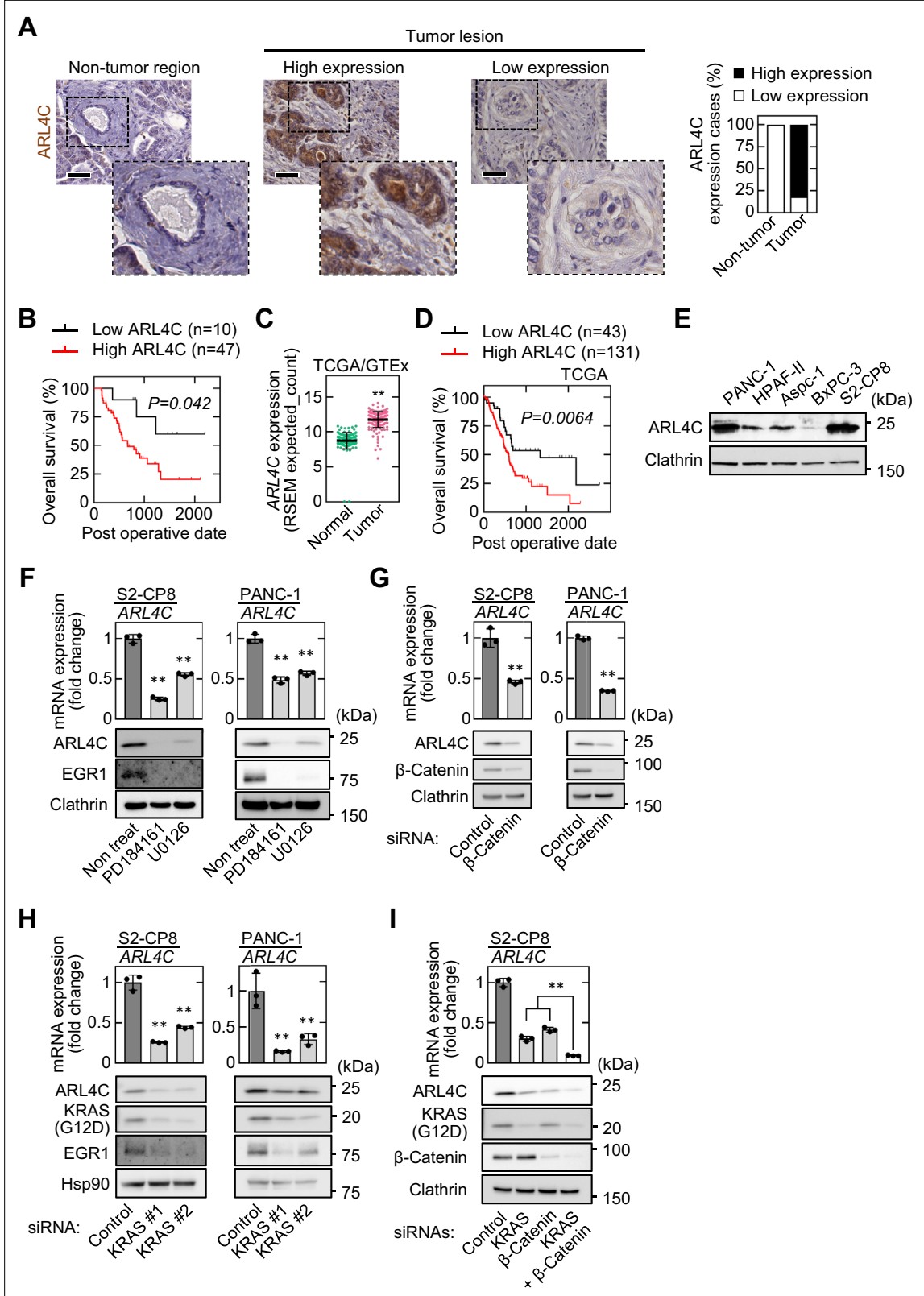

**Figure 1.** ARL4C is expressed in human pancreatic cancer. (**A**) PDAC tissues (n = 57) were stained with anti-ARL4C antibody and hematoxylin. The percentages of ARL4C expression cases in the non-tumor regions and tumor lesions are shown. (**B**) The relationship between overall survival and ARL4C expression in patients with PDAC. (**C**) *ARL4C* mRNA levels in pancreatic adenocarcinoma and normal pancreatic tissues were analyzed using TCGA and GTEx datasets. The results shown are scatter plots with the mean ± s.e.m. p Values were calculated using a two-tailed Student's t-test. (**D**) TCGA

*Figure 1 continued on next page*

*Figure 1 continued*

RNA sequencing and clinical outcome data for pancreatic cancer were analyzed. (**E**) Lysates of the indicated pancreatic cancer cells were probed with the indicated antibodies. (**F**) S2-CP8 and PANC-1 cells were treated with 10 μM PD184161 or 10 μM U0126, and *ARL4C* mRNA levels were measured by quantitative real-time PCR. Relative *ARL4C* mRNA levels were normalized to those of *GAPDH* and expressed as fold changes compared with the levels in control cells. Lysates were probed with the indicated antibodies. (**G–I**) S2-CP8 cells and PANC-1 cells were transfected with the indicated siRNAs, and *ARL4C* mRNA levels were measured by quantitative real-time PCR. Relative *ARL4C* mRNA levels were normalized to those of *B2M* and expressed as fold changes compared with the levels in control cells. Lysates were probed with the indicated antibodies. EGR1 was used as an established transcription target gene of RAS signaling. (**B,D**) Data were analyzed using Kaplan–Meier survival curves, and a log-rank test was used for statistical analysis. (**F–I**) Data are shown as the mean ± s.d. of three biological replicates. p Values were calculated using a two-tailed Student's t-test (**G**) or one-way ANOVA followed by Bonferroni post hoc test (**F,H,I**). Scale bars in (**A**) 50 μm. **, p < 0.01. See *Figure 1—source data 1*.

The online version of this article includes the following figure supplement(s) for figure 1:

**Source data 1.** Excel file containing quantitative data for *Figure 1*.

**Figure supplement 1.** ARL4C is expressed in pancreatic cancer cells.

A significant difference was observed between low and high ARL4C expression based on perineural invasion (*Supplementary file 1 table 1*). Because the perineural invasion is considered as one of the causes of the recurrence and metastasis after pancreatic resection (*Liang et al., 2016*), ARL4C expression may be correlated with the ability of cancer cell invasion. Consistently, ARL4C expression was correlated with decreased overall survival (*Figure 1B*). Analysis of TCGA and GTEx datasets revealed that ARL4C is highly expressed in tumor tissue than in non-diseased tissue (*Figure 1C*). In addition, when *ARL4C* high and low expression groups were separated based on the top 75 % of mRNA values of *ARL4C* in TCGA dataset, high expression of ARL4C indicated a poor prognosis (*Figure 1D*). Univariate and multivariate analysis revealed that higher ARL4C expression is an independent prognostic

**Table 1.** Univariate analysis and multivariate analysis of overall survival by Cox's Proportional Hazard model.

**Univariate analysis**

| Parameters | Hazard ratio | 95% CI | | P value |
|---|---|---|---|---|
| ARL4C(low/high) | 3.51 | 1.06 | 11.70 | 0.040 |
| Sex(Male/Female) | 1.10 | 0.54 | 2.24 | 0.80 |
| Age( < 65/≧65) | 1.05 | 0.47 | 2.35 | 0.91 |
| Tumor Location(Head/Body or Tail) | 0.41 | 0.18 | 0.94 | 0.036 |
| pStage(IA-IIA/IIB-III) | 2.51 | 1.17 | 5.41 | 0.019 |
| pT(1-2/3) | 5.29 | 1.23 | 22.70 | 0.025 |
| pN(0/1) | 2.51 | 1.17 | 5.41 | 0.019 |
| ly(0/1–3) | 2.74 | 1.17 | 6.46 | 0.021 |
| v(0/1–3) | 2.05 | 1.00 | 4.20 | 0.049 |
| ne(0/1–3) | 28,258 | 5.25E-36 | 1.52E + 44 | 0.83 |

**Multivariate analysis**

| Parameters | Hazard ratio | 95% CI | | P value |
|---|---|---|---|---|
| pT(1-2/3) | 3.72 | 0.78 | 17.7 | 0.099 |
| pN(0/1) | 1.80 | 0.79 | 4.10 | 0.16 |
| ARL4C(low/high) | 3.56 | 1.03 | 12.3 | 0.044 |

Hazard ratios with 95 % confidence intervals (CIs) were calculated using a Cox regression model and *P* values were calculated using a log-rank test. CI, confidence interval; pT, primary tumor; pN, regional lymph node; ly, lymphatic invasion; v, venous invasion; ne, perineural invasion.

factor (*Table 1*). Taken together, these results indicate that high expression of ARL4C is correlated with the aggressiveness and poor prognosis of pancreatic cancer.

Pancreatic intraepithelial neoplasia (PanIN) lesions were observed in 26 specimens. ARL4C was expressed in 20 of 26 cases (77%) of PanIN, suggesting that ARL4C is expressed in early stages of PDAC (*Figure 1—figure supplement 1C*). The results are consistent with our recent observations that ARL4C is frequently expressed in atypical adenomatous hyperplasia, which is the possible precursor lesions and develops to lung adenocarcinoma (*Kimura et al., 2020*).

In cultured pancreatic cancer cell lines, ARL4C was highly expressed in PANC-1 and S2-CP8 cells and it was barely detected in BxPC-3 cells (*Figure 1E*). Consistent with the previous results with IEC6 rat intestinal epithelial cells and colorectal and lung cancer cells (*Fujii et al., 2015*; *Matsumoto et al., 2014*), the MEK inhibitors PD184161 and U0126 and siRNAs for β-catenin and KRAS decreased ARL4C expression in S2-CP8 and PANC-1 cells (*Figure 1F–H*). In addition, simultaneous knockdown of KRAS and β-catenin further suppressed ARL4C expression (*Figure 1I*). Taken together, these results suggest that ARL4C is expressed in pancreatic cancer cells through activated RAS–MAP kinase and Wnt–β-catenin pathways.

## ARL4C expression is involved in the invasion of pancreatic cancer cells

ARL4C ASO-1316 has been shown to inhibit growth of xenograft tumors induced by colon and lung cancer cells (*Harada et al., 2019*; *Kimura et al., 2020*). However, ARL4C ASO-1316 had little effect on sphere formation of pancreatic cancer cell (*Figure 2—figure supplement 1A and B* and B) and did not induce cell death, which is assessed by propidium iodide (PI) staining (*Figure 2—figure supplement 1C*). Since the clinicopathological analysis of human pancreatic cancer specimens indicates that ARL4C expression may be correlated with invasive ability, migratory and invasive abilities of S2-CP8 and PANC-1 cells were studied in Boyden chamber assays. ARL4C ASO-1316 inhibited the migratory and invasive abilities with dominant effects on invasion (*Figure 2A and B*; *Figure 2—figure supplement 1D*). Inhibition of migratory and invasive abilities by ARL4C ASO, targeting the non-coding region of *ARL4C* mRNA, was not observed in the cells expressing ARL4C-GFP ectopically (*Figure 2C and D*; *Figure 2—figure supplement 1E*). Thus, ARL4C could be involved in migration and invasion of pancreatic cancer cells.

ARL4C has been shown to be localized to membrane protrusions of non-tumor cells, such as IEC6 and Madin-Darby canine kidney (MDCK) cells (*Matsumoto et al., 2014*). ARL4C-tdTomato was localized to protrusive structures extending from S2-CP8 cells under Matrigel-coated 2D culture conditions (*Figure 2—figure supplement 1F*). At the structures, focal adhesion proteins such as paxillin, phosphorylated paxillin, FAK, and phosphorylated FAK were localized with ARL4C-tdTomato, also with F-actin (*Figure 2—figure supplement 1F*). Therefore, we defined the membrane protrusions as actin-based structures that contain the adhesion sites, of which length is longer than 10 μm and diameter is shorter than 10 μm. ARL4C is unique in that it is locked to the GTP-bound active form, and ARL4C$^{Q72L}$-GFP, in which the amino acid at the same position in a constitutively active RAS mutant was mutated, showed a similar distribution to ARL4C-GFP. However, ARL4C$^{T27N}$-GFP, which is an inactive form (*Hofmann et al., 2007*), was not present in the protrusions (*Figure 2—figure supplement 1G*). These results suggest that ARL4C is present in the tips of membrane protrusions where it is expressed as wild type.

Invadopodia are well-known membrane protrusions that localize at the ventral surfaces of cells and are active in extracellular matrix (ECM) degradation during cancer invasion (*Murphy and Courtneidge, 2011*). To analyze invadopodia, pancreatic cancer cells were grown on gelatin-coated glass coverslips (*Figure 2—figure supplement 2A*). Dark areas represent gelatinolytic activity of invadopodia and are equal to invadopodia structures. BxPC-3 cells exhibited invadopodia clearly, whereas S2-CP8 and PANC-1 cells did not (*Figure 2—figure supplement 2A*). Thus, some pancreatic cancer cells do not form typical invadopodia in gelatin surface but can invade into ECM through probably other structures. Meanwhile, components of invadopodia, such as cortactin and ARPC2, were localized at the tips of protrusions defined above, with ARL4C (*Figure 2—figure supplement 2B*), suggesting that the protrusions might contribute to invasive phenotypes of pancreatic cancer cells and ARL4C functions there. Therefore, we referred to the protrusive structures as 'invasive pseudopods', because they seem to be analogous to invadopodia (*Jacquemet et al., 2013*; *Murphy and Courtneidge, 2011*; *Yu and Machesky, 2012*).

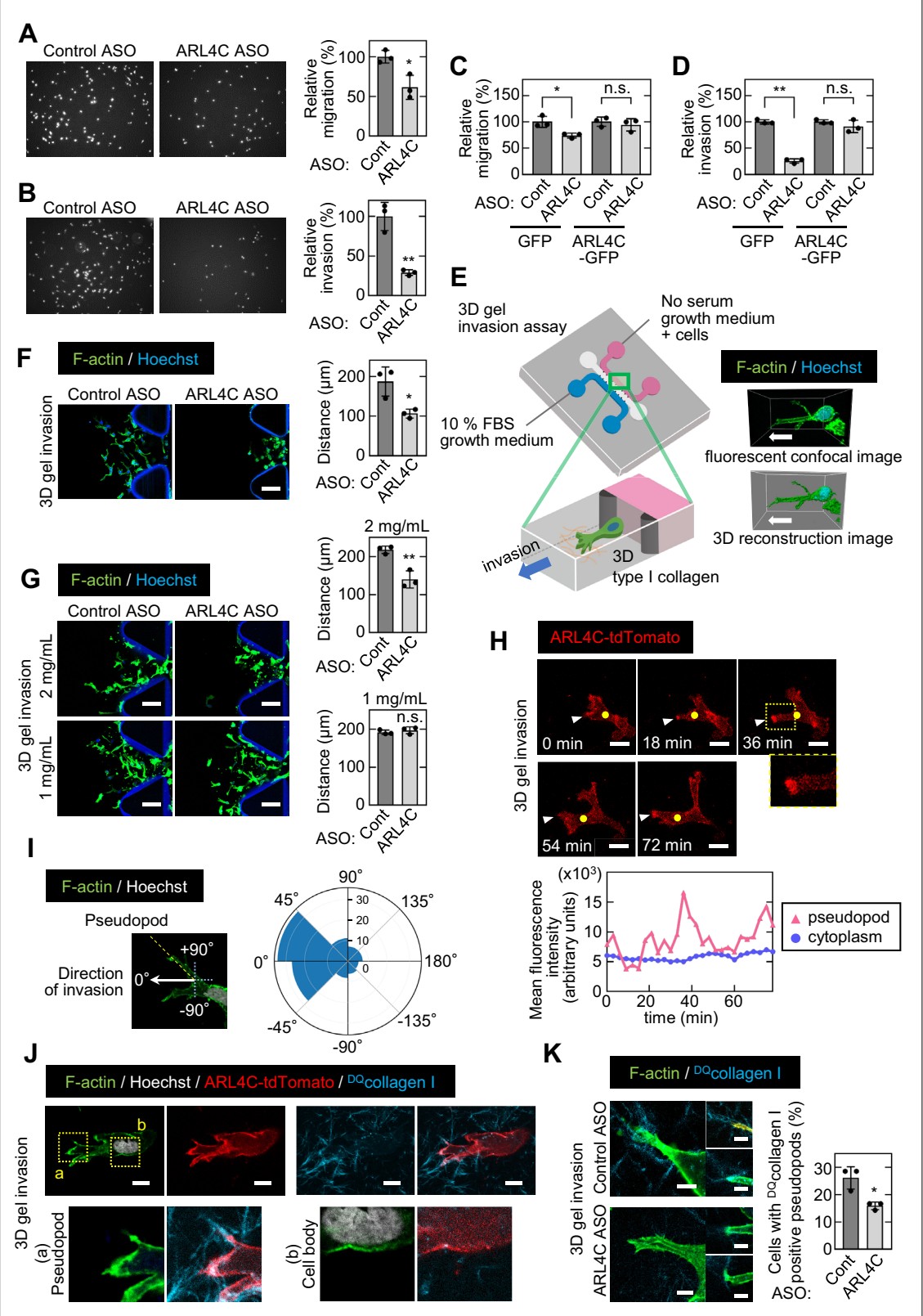

**Figure 2.** ARL4C expression is especially involved in the invasion of pancreatic cancer cells. (**A-D**) S2-CP8 cells (**A,B**) or S2-CP8 cells expressing GFP or ARL4C-GFP (**C,D**) were transfected with control or ARL4C ASO-1316 and subjected to migration (**A,C**) and invasion (**B,D**) assays. Migratory and invasive abilities are expressed as the percentage of the same cells transfected with control ASO. (**E**) A schematic illustration of 3D invasion into collagen I gel using a 3D cell culture chip is shown. There is a chemical concentration gradient across the gel channel and cells can invade into the gel. The

*Figure 2 continued on next page*

*Figure 2 continued*

right panel shows a fluorescent confocal image (top) and a 3D reconstructed image (bottom). (**F**) S2-CP8 cells were transfected with control or ARL4C ASO-1316 and subjected to a 3D collagen I gel (2 mg/mL) invasion assay. The distances from the edge of the gel interface of all cells invading into the collagen gel were measured. (**G**) The same assay as in (**F**) was performed in the presence of different concentrations of collagen I. (**H**) S2-CP8 cells stably expressing ARL4C-tdTomato were observed with time-lapse imaging. Arrowheads indicate the tips of invasive pseudopods and yellow circles indicate the cytoplasm (20 μm away from the tip of pseudopods). The region in the yellow dashed squares is shown enlarged in the bottom image. Fluorescence intensities of the cytoplasm and invasive pseudopods were measured and plotted as a function of time. (**I**) S2-CP8 cells were subjected to a 3D collagen I gel invasion assay and stained with phalloidin and Hoechst 33342. The angle of pseudopods to the direction of cell invasion toward FBS was calculated (n = 105). The results were plotted to a polar histogram. (**J**) S2-CP8 cells expressing ARL4C-tdTomato were subjected to a 3D collagen I gel invasion assay with $^{DQ}$collagen I, and stained with phalloidin and Hoechst 33342. The regions in the yellow dashed squares (a, pseudopod; b, cell body) are enlarged. (**K**) S2-CP8 cells transfected with control ASO or ARL4C ASO-1316 were subjected to a 3D collagen I gel invasion assay with $^{DQ}$collagen I. The percentages of cells with $^{DQ}$collagen I-positive pseudopods compared with the total number of cells were calculated. (**A–D,F,G,K**) Data are shown as the mean ± s.d. of three biological replicates. p Values were calculated using a two-tailed Student's t-test. Scale bars in (**F,G**) 100 μm; (**H**) 20 μm; (**J**) 10 μm; (**K**) 5 μm. n.s. not significant. *, p < 0.05; **, p < 0.01. See *Figure 2—source data 1*.

The online version of this article includes the following video and figure supplement(s) for figure 2:

**Source data 1.** Excel file containing quantitative data for *Figure 2*.

**Figure supplement 1.** ARL4C expression is involved in invasion of pancreatic cancer cells rather than in sphere formation.

**Figure supplement 1—source data 1.** Excel file containing quantitative data for *Figure 2—figure supplement 1*.

**Figure supplement 2.** ARL4C localizes at the tips of invasive pseudopods.

**Figure supplement 2—source data 1.** Excel file containing quantitative data for *Figure 2—figure supplement 2*.

**Figure supplement 3.** ARL4C is involved in invasion into 3D matrix.

**Figure supplement 3—source data 1.** Excel file containing quantitative data for *Figure 2—figure supplement 2*.

**Figure 2—video 1.** ARL4C accumulates at the tips of invasive pseudopods.

https://elifesciences.org/articles/66721/figures#fig2video1

**Figure 2—video 2.** S2-CP8 cells extend pseudopods to the direction of cell invasion.

https://elifesciences.org/articles/66721/figures#fig2video2

ARL4C knockout did decrease numbers of invasive pseudopods but slightly, while knockdown of ARPC2, which regulates formation of pseudopods as one of the components of Arp2/3 complex, clearly reduced the number of pseudopods (*Figure 2—figure supplement 2C-G*). Next it was tested whether ARL4C is involved in the presence of invadopodia markers in the tips of pseudopods. ARL4C knockout did not affect ARPC2 staining statistically and reduced the staining of cortactin only modestly (*Figure 2—figure supplement 2H and I*). It is quite likely that ARL4C contributes to invasive properties through other than the formation of pseudopods. Therefore, ARL4C may be necessary for functions of invasive pseudopods rather than their formation. This prompted us to look further into the invading process.

For visualization of cancer cells invading through the ECM (*Poincloux et al., 2009*), a 3D microfluidic cell culture with type I collagen (*Farahat et al., 2012*; *Shin et al., 2012*) (3D gel invasion assay) was performed (*Figure 2E*). At 0 time the same numbers of cells treated with control and ARL4C ASO were placed in the starting position (*Figure 2—figure supplement 3A*), and after 72 hr directional invading ability was compared. Whereas control S2-CP8 cells invaded into type I collagen, ARL4C ASO decreased invasive ability (*Figure 2F*). ARL4C KO cells also decreased invasive ability (*Figure 2—figure supplement 3B*). When the collagen concentration was reduced, S2-CP8 cells invaded irrespective of ARL4C knockdown (*Figure 2G*), suggesting that their invasive ability is not required for cells to move into the ECM when collagen fiber-formed 3D net structures are sparse. Furthermore, in the 3D gel invasion assay ARL4C-tdTomato accumulated in the tips of invasive pseudopods (*Figure 2H*; *Figure 2—video 1*). Fluorescence intensities of ARL4C-tdTomato in the edges of the pseudopods and cytoplasm (20 μm away from the tip of pseudopod), respectively, were measured over time, and then the intensities were plotted as a function of time. The results indicate that ARL4C dynamically appeared and disappeared in the pseudopods, but it did not accumulate in the cytoplasm (*Figure 2H*). Using time-lapse imaging the angle of pseudopods to the direction of cell movement towards FBS was observed. Most of them were located in the angle of –45 to +45 degrees in the polar histogram, suggesting that invasive pseudopods play a role in purposeful directional invasion (*Figure 2I*; *Figure 2—video 2*).

To visualize the relationship between the localization of ARL4C and matrix degradation, the steady-state activity of cell-derived protease was measured as the dequenched signal emitted from collagen I fibers with dye-quenched (DQ) FITC (DQcollagen I) (*Wolf et al., 2007*) in the 3D gel invasion assay. Protease-induced fluorescence dequenching was detected in the collagen fibers crossing the tips of the pseudopods but not in the cell body (*Figure 2J*). Protease activity was decreased when ARL4C was depleted (*Figure 2K*), suggesting that ARL4C is involved in degradation of the ECM through its localization to the tips of invasive pseudopods and plays an important role in the invasion of pancreatic cancer cells.

## IQGAP1 is an ARL4C-interacting protein

ARL4C recruits cytohesin2 to the plasma membrane through their direct interaction in HeLa cells (*Hofmann et al., 2007*). In S2-CP8 cells, ARL4C did not bind to cytohesin2 (*Figure 3—figure supplement 1A*), and knockdown of cytohesin2 had no effect on the migratory or invasive ability (*Figure 3—figure supplement 1B*). Furthermore, cytohesin2 was distributed throughout the cytosol in S2-CP8 cells, whereas it was localized to the cell periphery of HeLaS3 cells (*Figure 3—figure supplement 1C*). Whereas ARL4C ASO inhibited RAC1 activity in A549 cells (*Fujii et al., 2015*), the ASO did not affect RAC1 activity in S2-CP8 cells (*Figure 3—figure supplement 1D*). Although ARL4C induces the nuclear import of YAP/TAZ in HCT116 cells (*Harada et al., 2019*), ARL4C knockdown did not inhibit it in pancreatic cancer cells (*Figure 3—figure supplement 1E*). These results suggest that cytohesin2 neither functions downstream of ARL4C nor is involved in migration or invasion of pancreatic cancer cells and prompted us to explore an uncharacterized effector protein of ARL4C.

ARL4C-FLAG-HA–binding proteins were precipitated and the precipitates were analyzed by mass spectrometry (*Figure 3A*). Among the possible interacting proteins, IQGAP1 was further studied (*Figure 3A*; *Supplementary file 1 table 2*; *Source data 2*) because its expression is associated with the aggressiveness of various types of cancer (*Johnson et al., 2009*). Ectopically expressed and endogenous ARL4C were associated with endogenous IQGAP1 in S2-CP8 cells (*Figure 3B and C*). ARL4C-FLAG-HA and ARL4C$^{Q72L}$-FLAG-HA formed a complex with GFP-IQGAP1 to the similar levels, but ARL4C$^{T27N}$-FLAG-HA showed diminished binding to GFP-IQGAP1 in X293T cells (*Figure 3D*).

Using another anti-ARL4C antibody for the immunocytochemical study (*Figure 3—figure supplement 2A and B*), ARL4C and IQGAP1 were shown to accumulate to invasive pseudopods at endogenous level in S2-CP8 and PANC-1 cells under Matrigel-coated 2D culture conditions (*Figure 3E*; *Figure 3—figure supplement 2C*). Colocalization of ARL4C and IQGAP1 at invasive pseudopods was observed in 94 % of cells with ARL4C accumulation to the pseudopods. In 3D culture conditions, IQGAP1 was found at the tips of invasive pseudopods, similar to ARL4C-tdTomato (*Figure 3F*). IQGAP1 siRNA inhibited the migratory and invasive abilities in S2-CP8 and PANC-1 cells, and the cells expressing GFP-IQGAP1 were resistant to IQGAP1 siRNA (*Figure 3G and H*; *Figure 3—figure supplement 2D and E*). Simultaneous knockdown of ARL4C and IQGAP1 decreased the invasive ability, but the inhibitory degree was similar to that induced by knockdown of either ARL4C or IQGAP1 (*Figure 3I*). Thus, IQGAP1 and ARL4C regulate invasion in identical signaling pathways.

IQGAP1 was highly expressed in 31 of 57 PDAC patients (54%) (*Figure 3J*). The anti-IQGAP1 antibody was validated by Western blotting and immunocytochemical and immunohistochemical analyses (*Figure 3—figure supplement 2E-G*). Although higher expression of IQGAP1 was not associated with clinical parameters (*Supplementary file 1 table 3*), IQGAP1 expression correlated with decreased overall survival (*Figure 3K*). Similar results were obtained from the analysis of TCGA and GTEx datasets (*Figure 3—figure supplement 2H and I*). TCGA dataset revealed that expression of *ARL4C* mRNA in pancreatic cancer patients is positively correlated with that of *IQGAP1* mRNA (*Figure 3L*). Of 47 PDAC patients with high ARL4C expression, IQGAP1 was highly expressed in 27 patients (*Supplementary file 1 table 4*). Higher expression of ARL4C in the patients positive for IQGAP1 was associated with perineural invasion (*Supplementary file 1 table 4*). The overall survival of the patients who were double positive for ARL4C and IQGAP1 tended to be worse although it is not statistically significant (*Figure 3—figure supplement 2J*). Therefore, the relationship between ARL4C and IQGAP1 expression on patient survival using public datasets was analyzed. Overall survival was significantly decreased in the order of low ARL4C/low IQGAP1, high ARL4C/low IQGAP1, and high ARL4C/high IQGAP1, although the result of low ARL4C/high IQGAP1 could not conclude because of the small case numbers (n = 2) (*Figure 3—figure supplement 2K*).

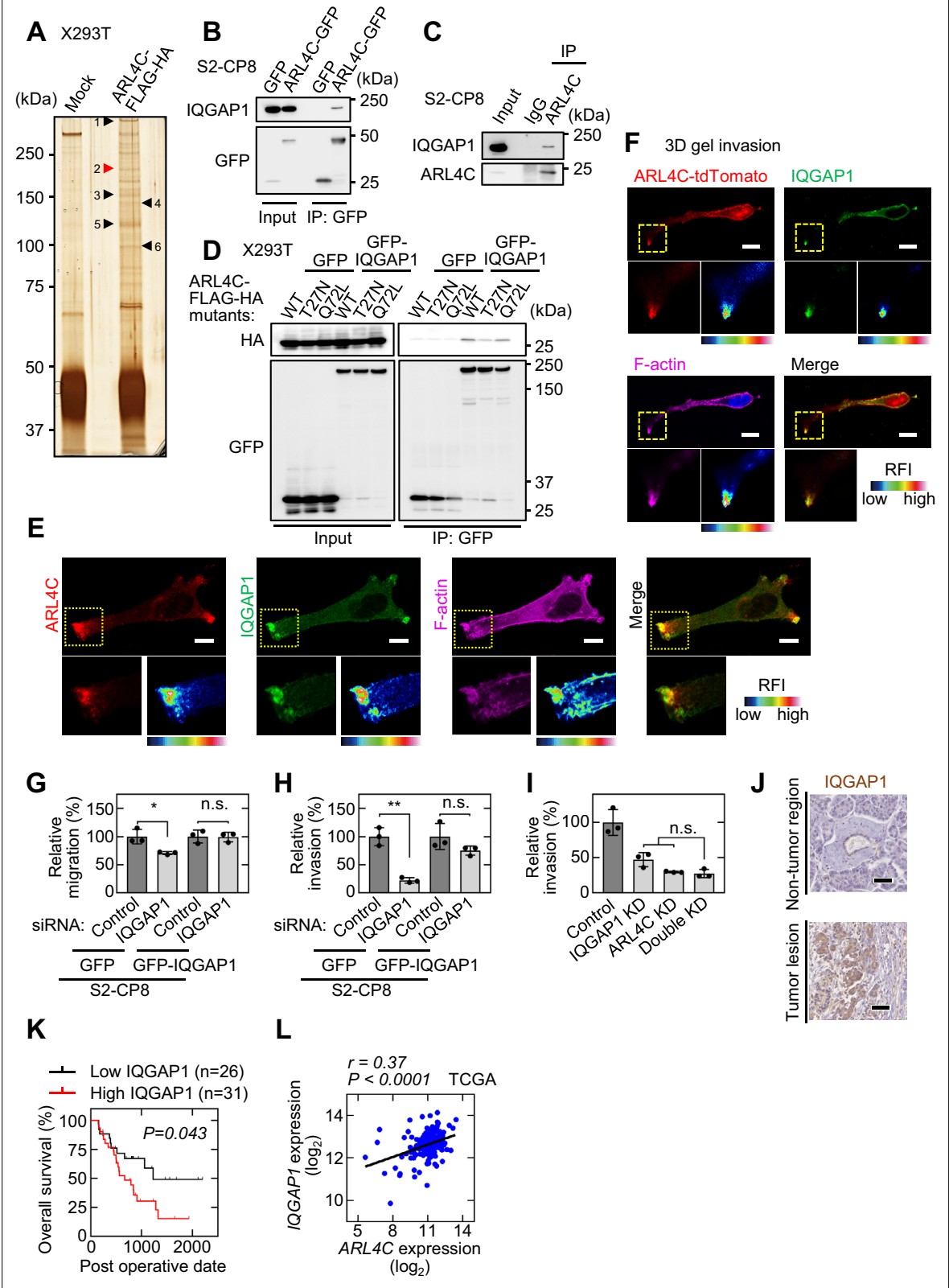

**Figure 3.** IQGAP1 is a novel ARL4C-interacting protein. (**A**) The ARL4C-interacting proteins in X293T cells were analyzed by mass spectrometry. The results are listed in *Supplementary file 1 table 2* and *Source data 2*. Arrowheads indicate the identified proteins, including IQGAP1 (red). (**B,C**) Lysates of S2-CP8 cells expressing ARL4C-GFP (**B**) or S2-CP8 WT cells (**C**) were immunoprecipitated with anti-GFP antibody (**B**) or anti-ARL4C antibody (**C**), and the immunoprecipitates were probed with the indicated antibodies. (**D**) Lysates of X293T cells expressing the indicated proteins were

*Figure 3 continued on next page*

*Figure 3 continued*

immunoprecipitated with anti-GFP antibody, and the immunoprecipitates were probed with the indicated antibodies. (**E**) S2-CP8 cells were stained with the indicated antibodies. Images of ARL4C and IQGAP1 were merged. (**F**) S2-CP8 cells expressing ARL4C-tdTomato were subjected to a 3D collagen I gel invasion assay and were stained with the indicated antibodies. Images of ARL4C and IQGAP1 were merged. (**G,H**) S2-CP8 cells expressing GFP or GFP-IQGAP1 were transfected with the indicated siRNAs and subjected to migration (**G**) and invasion (**H**) assays. Migratory and invasive abilities are expressed as the percentage of the same cells transfected with control siRNA. (**I**) S2-CP8 cells depleted of the indicated proteins were subjected to an invasion assay. Invasive activities are expressed as the percentage of control cells. (**J**) PDAC tissues were stained with anti-IQGAP1 antibody and hematoxylin. (**K**) The relationship between overall survival and IQGAP1 expression in PDAC patients was analyzed. (**L**) Scatter plot showing the correlation between the mRNA expression levels of *ARL4C* (X-axis) and *IQGAP1* (Y-axis) in pancreatic cancer patients obtained from TCGA datasets using the R2: Genomics Analysis and Visualization Platform. *r* indicates the Pearson's correlation coefficient. (**G-I**) Data are shown as the mean ± s.d. of three biological replicates. p Values were calculated using a two-tailed Student's t-test (**G,H**) or one-way ANOVA followed by Bonferroni post hoc test (**I**). (**K**) The data were analyzed by Kaplan–Meier survival curves, and a log-rank test was used for statistical analysis. (**E,F**) The regions in the yellow dashed squares are shown enlarged in the left bottom images. The right bottom images are shown with a false color representation of fluorescence intensity. More than 50 cells were imaged and the representative image is shown. False color representations were color-coded on the spectrum. Scale bars in (**E**) 10 µm; (**F**) 20 µm; (**J**) 50 µm. KD, knockdown. RFI, relative fluorescence intensity. n.s., not significant. *, p < 0.05; **, p < 0.01. See *Figure 3—source data 1*.

The online version of this article includes the following figure supplement(s) for figure 3:

**Source data 1.** Excel file containing quantitative data for *Figure 3*.

**Figure supplement 1.** Cytohesin2 does not mediate ARL4C signaling in pancreatic cancer cells.

**Figure supplement 1—source data 1.** Excel file containing quantitative data for *Figure 3—figure supplement 1*.

**Figure supplement 2.** IQGAP1 interacts with ARL4C and involves in the invasion of pancreatic cancer cells.

**Figure supplement 2—source data 1.** Excel file containing quantitative data for *Figure 3—figure supplement 2*.

Thus, simultaneous expression of ARL4C and IQGAP1 would be correlated with aggressiveness of pancreatic cancer.

## The polybasic region of ARL4C is required for its binding to IQGAP1

ARL4C is modified by myristate at the N terminus and has a polybasic region (PBR), comprising nine Lys or Arg residues, at the C terminus (*Donaldson and Jackson, 2011*). ARL4C$^{G2A}$, whose N-terminal myristoylation site (Gly2) is mutated to Ala, and ARL4C$^{\Delta PBR}$ were expressed in S2-CP8 cells. In contrast to ARL4C-GFP, ARL4C$^{G2A}$-GFP and ARL4C$^{\Delta PBR}$-GFP were not accumulated at invasive pseudopods where cortactin was present, but distributed throughout the cytosol (*Figure 4A and B*; *Figure 4—figure supplement 1A-C*), and both mutants severely decreased the binding activity to GFP-IQGAP1 (*Figure 4C*). The C-terminal region of KRAS includes the PBR and the CAAX motif, which is farnesylated, and fusion of the KRAS C-terminal region triggers the localization of the proteins to the cell surface membrane (*Hancock et al., 1990*). The KRAS C-terminal region was fused to the ARL4C mutants, which were referred to as ARL4C-GFP-Cterm. Both ARL4C$^{G2A}$-GFP-Cterm and ARL4C$^{\Delta PBR}$-GFP-Cterm were localized to invasive pseudopods where cortactin was present (*Figure 4B*; *Figure 4—figure supplement 1A*). However, although ARL4C$^{G2A}$-FLAG-HA-Cterm formed a complex with GFP-IQGAP1, ARL4C$^{\Delta PBR}$-FLAG-HA-Cterm did not (*Figure 4D*), suggesting that membrane localization of ARL4C is not sufficient for its binding to IQGAP1. Taken together, the PBR is necessary for ARL4C to associate with IQGAP1, as well as for recruiting ARL4C to invasive pseudopods.

The localization of IQGAP1 to invasive pseudopods was lost in ARL4C KO cells, but not vice versa (*Figure 4E and F*; *Figure 2—figure supplement 2C and D*; *Figure 4—figure supplement 1D-G*). The similar results were obtained in ARL4C knockdown cells (*Figure 4—figure supplement 1H*). In ARL4C KO cells, ARL4C-GFP and ARL4C$^{G2A}$-GFP-Cterm rescued the recruitment of IQGAP1 to the plasma membrane, unlike ARL4C$^{G2A}$-GFP, ARL4C$^{\Delta PBR}$-GFP, and ARL4C$^{\Delta PBR}$-GFP-Cterm (*Figure 4E*). Therefore, for IQGAP1 to be recruited to invasive pseudopods, the localization of ARL4C to the plasma membrane and the binding to IQGAP1 through the PBR might be necessary. In addition, inhibition of invasive ability by ARL4C ASO-1316 was cancelled by expression of ARL4C$^{G2A}$-GFP-Cterm but not by that of ARL4C$^{G2A}$-GFP, ARL4C$^{\Delta PBR}$-GFP, or ARL4C$^{\Delta PBR}$-GFP-Cterm (*Figure 4G*; *Figure 4—figure supplement 1I*). Thus, the binding of ARL4C and IQGAP1 in invasive pseudopods could be essential for the invasive ability.

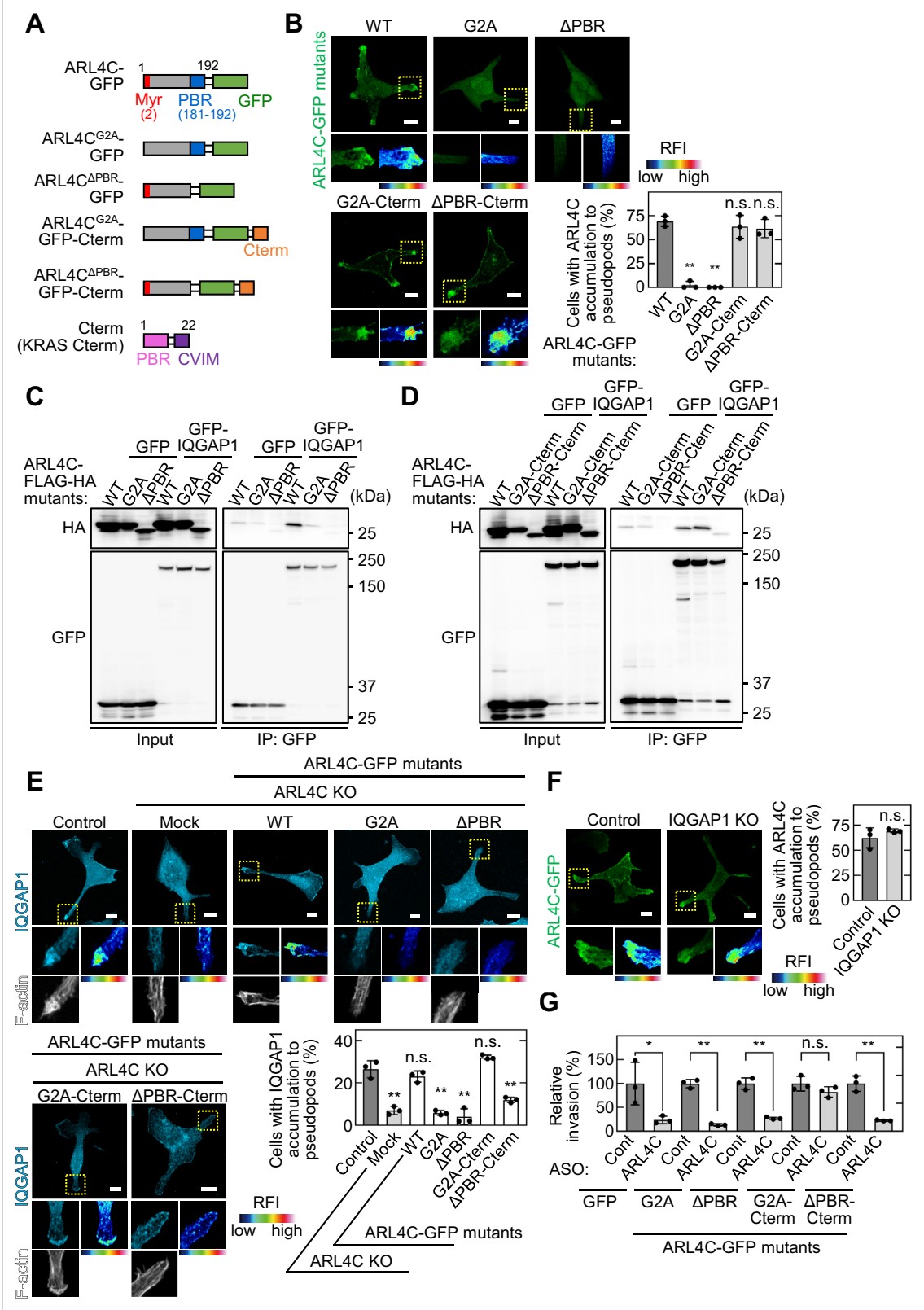

**Figure 4.** The PBR of ARL4C is required for ARL4C and IQGAP1 binding. (**A**) A schematic representation of four ARL4C-GFP mutants is shown. (**B**) S2-CP8 cells were transfected with the indicated mutants of ARL4C-GFP. The percentages of cells with ARL4C-GFP mutant accumulated at invasive pseudopods compared with the total number of cells were calculated. (**C,D**) Lysates of X293T cells expressing the indicated proteins were immunoprecipitated with anti-GFP antibody and the immunoprecipitates were probed with anti-HA and anti-GFP antibodies. (**E**) S2-CP8 WT or ARL4C

*Figure 4 continued on next page*

DOI: https://doi.org/10.7554/eLife.66721

**Figure 4 continued**

KO cells transfected with control or the indicated mutants of ARL4C-GFP were stained with anti-IQGAP1 antibody and phalloidin. The percentages of cells with IQGAP1 accumulated at invasive pseudopods compared with the total number of cells were calculated. (**F**) S2-CP8 WT or IQGAP1 KO cells were transfected with ARL4C-GFP. The percentages of cells with ARL4C-GFP accumulated at invasive pseudopods compared with the total number of cells were calculated. (**G**) S2-CP8 cells stably expressing GFP or the indicated mutants of ARL4C-GFP were transfected with control or ARL4C ASO and subjected to invasion assays. Invasive ability is expressed as the percentage of the same cells transfected with control ASO. (**B,E–G**) Data are shown as the mean ± s.d. of three biological replicates. p Values were calculated using a two-tailed Student's t-test (**F,G**) or one-way ANOVA followed by Bonferroni post hoc test (**B,E**). (**B,E,F**) The regions in the yellow dashed squares are shown enlarged in the left bottom images. The right bottom images are shown in a false color representation of fluorescence intensity. False color representations were color-coded on the spectrum. Scale bars in (**B,E,F**) 10 μm. KO, knockout. RFI, relative fluorescence intensity. n.s., not significant. *, p < 0.05; **, p < 0.01. See *Figure 4—source data 1*.

The online version of this article includes the following figure supplement(s) for figure 4:

**Source data 1.** Excel file containing quantitative data for *Figure 4*.

**Figure supplement 1.** ARL4C is essential for recruitment of IQGAP1 to invasive pseudopods.

**Figure supplement 1—source data 1.** Excel file containing quantitative data for *Figure 4—figure supplement 1*.

## ARL4C recruits IQGAP1 to invasive pseudopods in a PI(3,4,5)P3-dependent manner

PI(4,5)P2 (PIP2) and PI(3,4,5)P3 (PIP3) are required for ARL4C membrane targeting (*Heo et al., 2006*). The pleckstrin homology (PH) domain functions as a protein- and phospholipid-binding structural protein module (*Maffucci and Falasca, 2001*). The PH domains of PLCδ and GRP1 prefer to bind to PIP2 and PIP3, respectively (*Lemmon, 2008*). GFP-PLCδ$^{PH}$ was detected throughout the cell surface membrane, whereas GFP-GRP1$^{PH}$ was accumulated in invasive pseudopods (*Figure 5A*).

The levels of PIP2 and PIP3 in the plasma membrane were decreased by a rapamycin-inducible PIP2-specific phosphatase (Inp54p) (*Suh et al., 2006*) and a PI3 kinase inhibitor LY294002 (*Petrie et al., 2012*), respectively. S2-CP8 cells were treated with rapamycin and LY294002 for 30 min to examine the localization of ARL4C and IQGAP1, and for 24 hr to test invasive ability. PIP3 depletion decreased the membrane targeting of ARL4C and IQGAP1 and reduced the invasive ability, but PIP2 depletion did not (*Figure 5B and C*). IQGAP1 and ARL4C-mCherry colocalized with GRP1$^{PH}$ in invasive pseudopods (*Figure 5D*), suggesting that both proteins accumulate in the cell peripheral regions containing PIP3 and promote invasion.

To reveal the importance of PIP3 for the localization area of ARL4C and IQGAP1, PLCδ$^{PH}$ or GRP1$^{PH}$ was fused to the C terminus of ARL4C$^{G2A}$-GFP (*Figure 5E*). While both ARL4C$^{G2A}$-GFP-GRP1$^{PH}$ and ARL4C$^{G2A}$-GFP-PLCδ$^{PH}$ formed a complex with GFP-IQGAP1, the former construct was localized to invasive pseudopods, but the latter construct was present throughout the cell surface membrane (*Figure 5F*; *Figure 5—figure supplement 1A*). Consistently, in ARL4C KO cells extending invasive pseudopods, the localization of IQGAP1 to invasive pseudopods was rescued by ARL4C$^{G2A}$-GFP-GRP1$^{PH}$ but not by ARL4C$^{G2A}$-GFP-PLCδ$^{PH}$ (*Figure 5G*). Furthermore, ARL4C ASO-1316 inhibited the invasive ability of S2-CP8 cells expressing ARL4C$^{G2A}$-GFP-PLCδ$^{PH}$ but not those expressing ARL4C$^{G2A}$-GFP-GRP1$^{PH}$ (*Figure 5H*; *Figure 5—figure supplement 1B*). Taken together, these results suggest that PIP3-dependent membrane targeting of ARL4C recruits IQGAP1 to invasive pseudopods and promotes invasion.

## ARL4C is involved in the focal delivery of MMP14 to invasive pseudopods through IQGAP1

IQGAP1 is involved in the trafficking of MMP14-containing vesicles to the leading structures of cancer cells (*Sakurai-Yageta et al., 2008*). TCGA dataset showed that expression of *MMP14* mRNA in pancreatic cancer patients is positively correlated with that of both *ARL4C* and *IQGAP1* mRNA (*Figure 6—figure supplement 1A*). In addition, MMP14 expression was associated with poor prognosis (*Figure 6—figure supplement 1B*).

Cell surface MMP14-GFP accumulated in invasive pseudopods containing IQGAP1 and ARL4C-FLAG-HA (*Figure 6A*). MMP14-GFP extremely disappeared from invasive pseudopods of ARL4C knockdown and KO cells and the phenotype was rescued by expression of ARL4C-FLAG-HA (*Figure 6B*; *Figure 6—figure supplement 1C* and D). IQGAP1 KO caused the loss of MMP14-GFP from the pseudopods, and FLAG-HA-IQGAP1 expression rescued this phenotype (*Figure 6C*). The

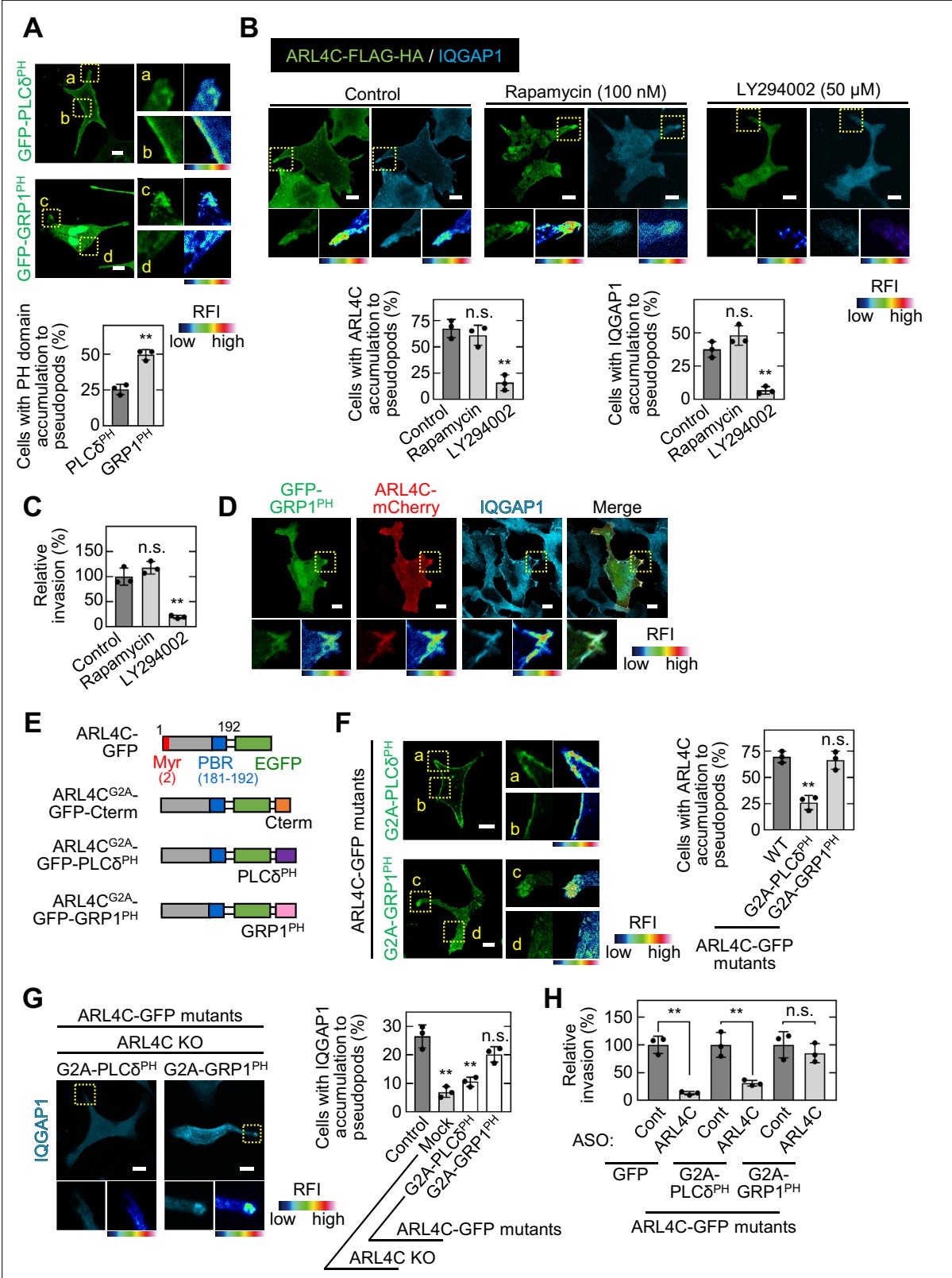

**Figure 5.** ARL4C recruits IQGAP1 to invasive pseudopods in a PIP3-dependent manner. (**A**) S2-CP8 cells were transfected with GFP-PLC δ $^{PH}$ or GFP-GRP1$^{PH}$. The percentages of cells with GFP-PLC δ $^{PH}$ or GFP-GRP1$^{PH}$ accumulated at invasive pseudopods compared with the total number of cells were calculated. (**B**) S2-CP8 cells expressing FRB-CFP, mRFP-FKBP-5-ptase domain, and ARL4C-FLAG-HA were treated with or without rapamycin or LY294002 and stained with anti-HA and anti-IQGAP1 antibodies. The percentages of cells with IQGAP1 or ARL4C-FLAG-HA accumulated at invasive pseudopods

*Figure 5 continued on next page*

*Figure 5 continued*

compared with the total number of cells were calculated. (**C**) S2-CP8 cells expressing FRB-CFP and mRFP-FKBP-5-ptase domain were treated with or without rapamycin or LY294002 and subjected to an invasion assay. Invasive abilities are expressed as the percentage of control cells. (**D**) S2-CP8 cells expressing ARL4C-mCherry and GFP-GRP1$^{PH}$ were stained with anti-IQGAP1 antibody. Images of GFP-GRP1$^{PH}$, ARL4C-mCherry, and IQGAP1 were merged. (**E**) A schematic representation of ARL4C-GFP mutants is shown. (**F**) S2-CP8 cells were transfected with the indicated mutants of ARL4C-GFP. The percentages of cells with ARL4C-GFP mutant accumulated at invasive pseudopods compared with the total number of cells were calculated. (**G**) ARL4C KO cells expressing control or the indicated mutants of ARL4C-GFP were stained with anti-IQGAP1 antibody. Quantification was performed as in (**B**). (**H**) S2-CP8 cells stably expressing GFP or the indicated mutants of ARL4C-GFP were transfected with control or ARL4C ASO and subjected to an invasion assay. Invasive abilities are expressed as the percentage of the same cells transfected with control ASO. (**A,F**) Enlarged images of the regions in the yellow dashed squares and a false color representation of fluorescence intensity are shown on the right. (a) and (c) show the pseudopods, and (b) and (d) show the cell body. (**B,D,G**) The regions in the yellow dashed squares are shown enlarged in the left bottom images. The right bottom images are shown in a false color representation of fluorescence intensity. (**A–C,F–H**) Data are shown as the mean ± s.d. of three biological replicates. p Values were calculated using a two-tailed Student's t-test (**A,H**) or one-way ANOVA followed by Bonferroni post hoc test (**B,C,F,G**). (**A,B,D,F,G**) False color representations were color-coded on the spectrum. Scale bars in (**A,B,D,F,G**) 10 µm. KO, knockout. RFI, relative fluorescence intensity. n.s., not significant. **, p < 0.01. See *Figure 5—source data 1*.

The online version of this article includes the following figure supplement(s) for figure 5:

**Source data 1.** Excel file containing quantitative data for *Figure 5*.

**Figure supplement 1.** Interaction of ARL4C mutants and IQGAP1.

failure of MMP14 membrane targeting in ARL4C KO cells was rescued by expression of ARL4C$^{G2A}$-FLAG-HA-Cterm but not by that of ARL4C$^{G2A}$-FLAG-HA, ARL4C$^{\Delta PBR}$-FLAG-HA, or ARL4C$^{\Delta PBR}$-FLAG-HA-Cterm (*Figure 6B*). In addition, PIP3 depletion, but not PIP2 depletion, suppressed the membrane localization of MMP14 (*Figure 6D*). Therefore, in co-operation with ARL4C and IQGAP1, MMP14 is likely to be trafficked to invasive pseudopods with PIP3 accumulation.

Consistent with these results, the inhibited invasive ability after double knockdown of ARL4C and MMP14 or IQGAP1 and MMP14 was similar to that seen after single knockdown of ARL4C, IQGAP1, or MMP14 (*Figure 6E*; *Figure 6—figure supplement 1E and F*). Knockdown of ARL4C, IQGAP1, or MMP14 decreased invasive ability in 3D microfluidic cell culture (*Figure 6F*) and the protease activity was also reduced (*Figure 6G*). Previous work has shown that MMP14$^{\Delta C}$(Δ563–582) lacking the cytoplasmic region fails to be endocytosed (*Jiang et al., 2001*). Here, MMP14$^{\Delta C}$ was retained in invasive pseudopods of ARL4C-KO cells (*Figure 6—figure supplement 1G*), and the ARL4C knockdown-mediated decreases in cell invasion and collagen degradation were rescued by MMP14$^{\Delta C}$ (*Figure 6H*; *Figure 6—figure supplement 1H and I*). Thus, ARL4C-dependent recruitment of MMP14 to invasive pseudopods is required for cell invasion.

Pancreatic cancer tissues were stained with anti-ARL4C, anti-IQGAP1, and anti-MMP14 antibodies in the serial section. Notably, ARL4C and MMP14 were expressed more highly in invasive cancer cells rather than in PanIN lesions, although IQGAP1 was thoroughly expressed in tumor lesions including PanIN area (*Figure 6I*). Using triple immunofluorescence imaging assay, it was confirmed that three proteins are simultaneously expressed in PDAC cells invading the surrounding interstitial tissues (*Figure 6—figure supplement 1J*). Taken together, these results support the idea that the ARL4C–IQGAP1–MMP14 signaling axis participates in pancreatic cancer cell invasion.

## ARL4C ASO inhibits pancreatic tumor metastasis in vivo

To show that ARL4C is indeed involved in cancer cell invasion in vivo, the effects of subcutaneous injection of ARL4C ASO-1316 on an orthotopic transplantation model were tested. S2-CP8 cells expressing luciferase were injected into the pancreas of nude mice, and control ASO or ARL4C ASO-1316 was subcutaneously injected from day 3 (*Figure 7A*). After 2 and 3 weeks, ARL4C ASO-1316 suppressed the luminescence signal compared with control ASO (*Figure 7B*), and ARL4C expression was decreased immunohistochemically (*Figure 7C*). Whereas ARL4C ASO-1316 did not reduce the size of the primary tumor in the pancreas, the ASO decreased the numbers of lymph node metastases and tended to improve the survival (*Figure 7D and E*; *Figure 7—figure supplement 1A*).

When 6-FAM–labeled ARL4C ASO-1316 was subcutaneously injected into tumor-bearing mice, the fluorescence was extremely detected in the pancreas and slightly observed in the kidney which is due to renal excretion (*Figure 7F*). 6-FAM–labeled ARL4C ASO-1316 was highly accumulated in tumor lesions but not in the neighboring normal tissues (*Figure 7G*), indicating that ASO was incorporated

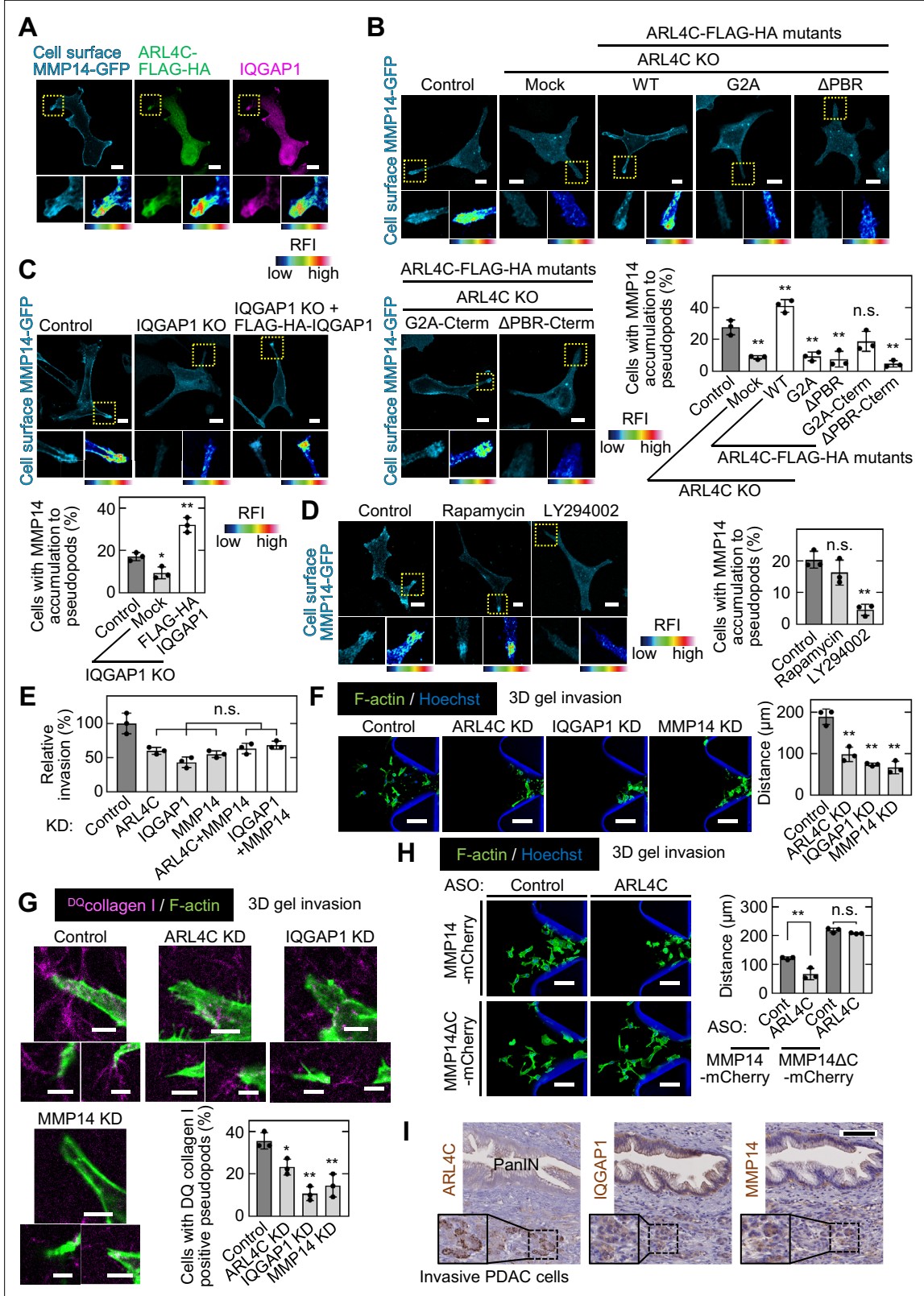

**Figure 6.** ARL4C is involved in focal delivery of MMP14 to invasive pseudopods through IQGAP1. (**A**) S2-CP8 cells expressing MMP14-GFP and ARL4C-FLAG-HA were stained with anti-MMP14 without permeabilization, followed by permeabilization and staining with anti-HA and anti-IQGAP1 antibodies. (**B**) S2-CP8 WT or ARL4C KO cells expressing MMP14-GFP and the indicated mutants of ARL4C-FLAG-HA were stained with anti-MMP14 without permeabilization. The percentages of cells with MMP14 accumulated at invasive pseudopods compared with the total number of cells were

*Figure 6 continued on next page*

*Figure 6 continued*

calculated. (**C**) The same assay as in (**B**) was performed except with S2-CP8 WT or IQGAP1 KO cells expressing MMP14-GFP and FLAG-HA-IQGAP1. (**D**) S2-CP8 cells expressing MMP14-GFP, FRB-CFP, and mRFP-FKBP-5-ptase domain were treated with 100 nM rapamycin or 50 µM LY294002 for 30 min. Staining and quantification were performed as in (**B**). (**E**) S2-CP8 cells depleted of the indicated proteins were subjected to an invasion assay. Invasive activities are expressed as the percentage of control cells. (**F–H**) S2-CP8 cells (**F,G**) or S2-CP8 cells expressing MMP14-mCherry or MMP14ΔC-mCherry (**H**) depleted of the indicated proteins were subjected to a 3D collagen I gel invasion assay with <sup>DQ</sup>collagen I. The distances from the edge of the gel interface of all cells that invaded into the gel were measured (**F,H**). The percentages of cells with <sup>DQ</sup>collagen I-positive pseudopods compared with the total number of cells were calculated (**G**). (**I**) PDAC tissues were stained with the indicated antibodies and hematoxylin. The regions in the black dashed squares are shown enlarged in the solid squares. Nine patient samples were imaged and the representative images are shown. (**A–D**) The regions in the yellow dashed squares are shown enlarged in left bottom and a false color representation of fluorescence intensity is shown in right bottom. False color representations were color-coded on the spectrum. (**B–H**) Data are shown as the mean ± s.d. of three biological replicates. p Values were calculated using a two-tailed Student's t-test (**H**) or one-way ANOVA followed by Bonferroni post hoc test (**B–G**). Scale bars in (**A–D**) 10 µm; (**F,H**) 100 µm; (**G**) 5 µm; (**I**) 100 µm. KO, knockout; KD, knockdown. RFI, relative fluorescence intensity. n.s., not significant. *, p < 0.05; **, p < 0.01. See *Figure 6—source data 1*.

The online version of this article includes the following figure supplement(s) for figure 6:

**Source data 1.** Excel file containing quantitative data for *Figure 6*.

**Figure supplement 1.** ARL4C recruits MMP14 to invasive pseudopods and their expression is associated with poor prognosis in pancreatic cancer patients.

**Figure supplement 1—source data 1.** Excel file containing quantitative data for *Figure 6—figure supplement 1*.

into tumor lesions after systemic injection. In primary pancreatic tumors, ARL4C ASO-1316 reduced ARL4C expression at protein and mRNA levels (*Figure 7H*; *Figure 7—figure supplement 1B*) and decreased the localization of IQGAP1 to the cell surface area (*Figure 7I*; *Figure 7—figure supplement 1C*). Tumor cells were observed in lymphatic vessels of peritumoral areas of control ASO-treated mice but not in those of ARL4C ASO-treated mice (*Figure 7J*). In addition, tumor cells surrounding peritumoral lymphatic vessels were also decreased, which is consistent with our hypothesis that ARL4C is required for cell invasive activity.

To compare molecular characteristics between pancreatic tumors from mice injected with control ASO and ARL4C ASO-1316, RNA sequence analysis was performed for primary tumors (*Figure 7—source data 2*). Principal component analysis (PCA) indicated a clear difference in the gene expression profiles of tumors from control ASO- and ARL4C ASO-1316–treated mice (*Figure 7K*). Furthermore, hierarchical clustering revealed a drastic change in expression of genes due to ARL4C ASO-1316 injection (*Figure 7L*). Two hundred and three differentially expressed genes (DEGs) were detected, and by subjecting them to Ingenuity Pathway Analysis (IPA), the top five significantly enriched terms of the biological process of molecular function in the inhibition and activation of the pathways were obtained (*Figure 7M*). In particular, DEGs linked to the inhibition of the pathways in ARL4C ASO-1316–treated mice were predicted to be involved in terms such as cell migration and invasion (*Figure 7M*). Taken together, these results suggest that ARL4C ASO inhibits the invasion of tumor cells into lymphatic vessels in vivo, and the gene profiles of tumors treated with ARL4C ASO in vivo support the putative functions of ARL4C in pancreatic cancer invasion.

## Discussion

Pancreatic cancer represents one of the leading causes of cancer death, despite advances in cancer therapy (*Keleg et al., 2003*). Major problem of pancreatic cancer is uncontrollable invasion and metastasis. In this study, we found that the ARL4C–IQGAP1–MMP14 signaling axis is involved in pancreatic cancer invasion. Because ARL4C expression is induced by Wnt and EGF signaling, it is reasonable that ARL4C would be expressed in a β-catenin- and RAS-dependent manner in pancreatic cancer cells. ARL4C is a unique small G protein because it is constitutively active, regardless of wild-type (*Burd et al., 2004*; *Matsumoto et al., 2017*). The long interswitch region of ARL4C may prevent the retractile conformation change in the GDP-bound state (*Burd et al., 2004*; *Pasqualato et al., 2002*). ARL4C could be a constitutively active form without active mutations, and its activity may be controlled by transcriptional regulation.

ARL4C binds to cytohesin2 (*Hofmann et al., 2007*), leading to activation of ARF6–RAC–RHO–YAP/TAZ signaling in colon and lung cancer cells (*Fujii et al., 2015*; *Kimura et al., 2020*). Because ARL4C did not bind to cytohesin2 but to IQGAP1 in pancreatic cancer cells, it is likely that ARL4C regulates

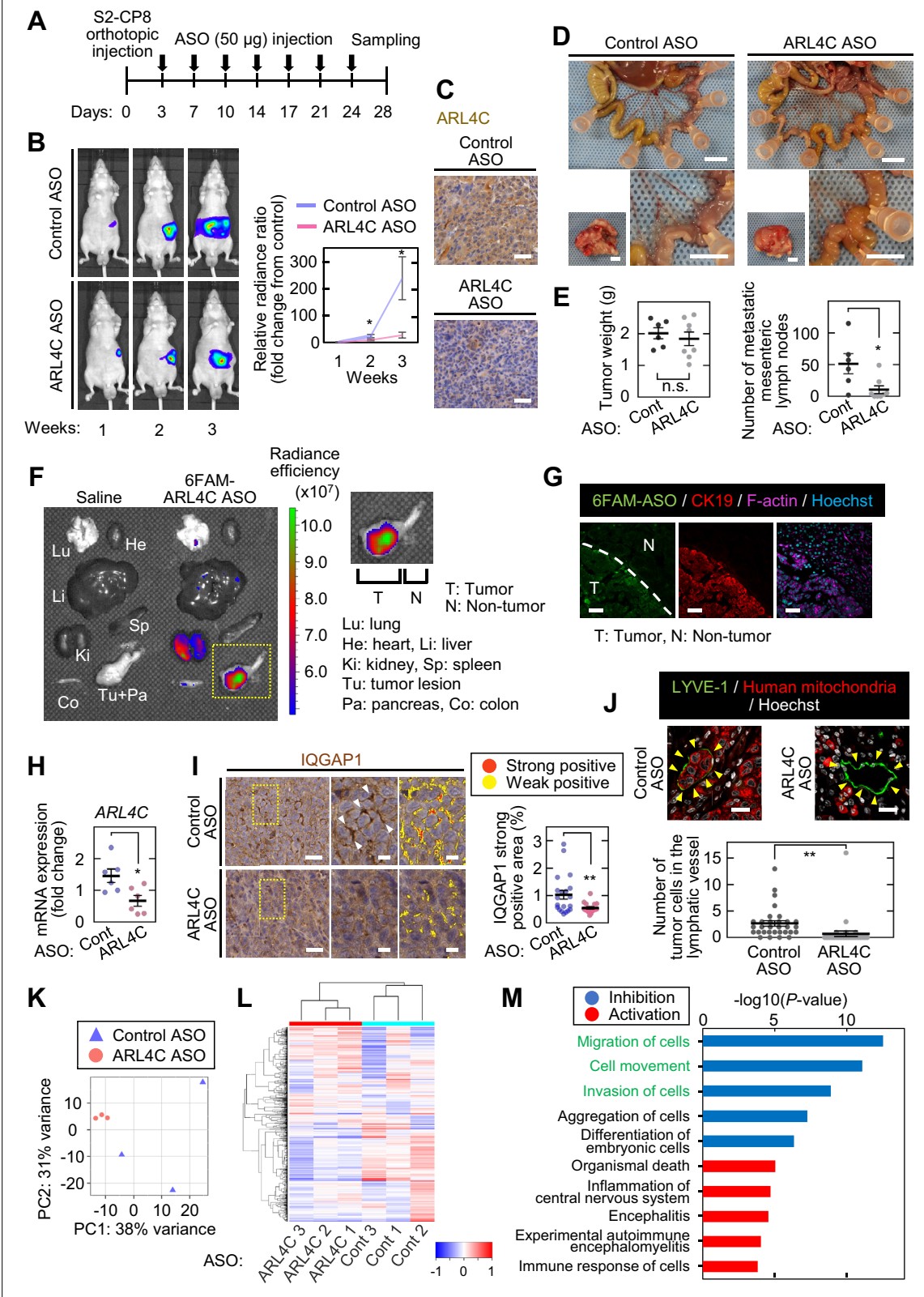

**Figure 7.** ARL4C ASO inhibits pancreatic tumor metastasis in vivo. (**A**) S2-CP8/Luciferase cells were implanted into the pancreas of nude mice, and control ASO (n = 6) or ARL4C ASO-1316 (n = 7) was subcutaneously administered. (**B**) Bioluminescence images of the intraperitoneal tumors are presented (left) and quantification of the tumor burden is shown (right). The data are presented as the mean ± s.e.m. of the fold change in luminescent intensity relative to that of week 1 treated with control ASO. (**C**) Sections from the pancreatic tumors from control ASO- or ARL4C ASO-1316-treated

*Figure 7 continued on next page*

*Figure 7 continued*

tumor-bearing mice were stained with anti-ARL4C antibody and hematoxylin. (**D**) Representative images of the tumors in the pancreas (left bottom) and metastatic mesenteric lymph nodes (top and right bottom) are shown. (**E**) Primary tumor weight (left) and metastatic mesenteric lymph node number are presented (right). Data are shown as the mean ± s.e.m. (**F,G**) Four hr after subcutaneous injection of 6-FAM-ARL4C ASO-1316 into tumor-bearing mice, the fluorescence intensities of various organs were measured (**F**), and the sections prepared from the pancreas were stained with the indicated antibodies (**G**). Area indicated by yellow dashed square is enlarged on the right panel (**F**). (**H**) Total RNA was extracted from tumors of control ASO- or ARL4C ASO-1316-treated tumor-bearing mice. Tumor block was cut into three pieces from each tumor of 2 mice per group. *ARL4C* mRNA levels were measured by quantitative real-time PCR. Relative *ARL4C* mRNA levels were normalized to those of *B2M* and expressed as fold changes compared with the levels in one of the control samples. (**I**) Sections from the pancreatic tumors were stained with anti-IQGAP1 antibody and hematoxylin. The two panels on the right show enlarged images of the yellow dashed squares. Positive staining of IQGAP1 is color-coded as yellow (weakly positive) or red (strongly positive). The percentage of the strongly positive IQGAP1 area was calculated. Data are shown as the mean ± s.e.m. Twenty fields were analyzed from 3 mice per group. (**J**) Sections from the pancreatic tumor were stained with the indicated antibodies. The numbers of tumor cells in the lymphatic vessels (indicated with yellow arrowheads) were counted. Data are shown as the mean ± s.e.m. Thirty lymphatic vessels were analyzed from 3 mice per group. (**K,L**) RNA sequencing was performed for S2-CP8-derived primary tumors, and the results of principal component analysis (**K**) and hierarchical clustering (**L**) are shown. (**M**) Differentially expressed genes were subjected to Ingenuity Pathway Analysis (IPA). The top five disease or function annotations of the positive and negative Z-score groups are shown. Bars indicate the -log10 (p value). Inhibited pathways are represented by blue-colored bars while activated pathways are shown by red-colored bars. (**B,E,H-J**) p Values were calculated using a two-tailed Student's t-test. Scale bars in (**D**) 5 mm; (**C,G,I**) 50 μm; (**J**) 20 μm. n.s., not significant. *, p < 0.05; **, p < 0.01. See *Figure 7—source data 1* and *Figure 7—source data 2*.

The online version of this article includes the following figure supplement(s) for figure 7:

**Source data 1.** Excel file containing quantitative data for *Figure 7*.

**Source data 2.** Excel file containing quantitative data for *Figure 7*.

**Figure supplement 1.** ARL4C ASO-1316 extends the survival of orthotopically transplanted mice.

**Figure supplement 1—source data 1.** Excel file containing quantitative data for *Figure 7—figure supplement 1*.

different downstream signaling pathways in a cancer cell context-dependent manner. Invadopodia are the unique structures observed at the ventral sites of certain types of cancer cells, such as BxPC-3, breast cancer MDA-MB-231 cells, and head and neck squamous carcinoma SCC61 cells (*Dalaka et al., 2020*; *Murphy and Courtneidge, 2011*), but the typical invadopodia are not observed in S2-CP8 and PANC-1 cells. Invasive pseudopods that we defined in these pancreatic cancer cells highly expressing ARL4C consisted of similar molecules, including cortactin, ARPC2, IQGAP1, and MMP14, which are involved in invadopodia functions (*Caswell and Zech, 2018*; *Jacquemet et al., 2013*; *Murphy and Courtneidge, 2011*). ARL4C depletion severely suppresses the localization of IQGAP1 and MMP14 to pseudopods and inhibits invasive ability, but does affect the localization of cortactin as well as the structure itself only moderately. Therefore, major function of ARL4C in invasive pseudopods would be to recruit MMP14 by binding to IQGAP1 rather than pseudopod formation.

Both myristoylation and the PBR of ARL4C support plasma membrane targeting (*Heo et al., 2006*). In our results, both motifs were necessary for the localization of ARL4C to the plasma membrane, whereas the PBR, rather than myristoylation, was indispensable for the activity of the ARL4C–IQGAP1–MMP14 signaling axis. Phosphoinositides have been implicated in many aspects of cell physiology (*Di Paolo and De Camilli, 2006*). PIP3 is localized to the leading edge of migrating cells and invadopodia of cancer cells (*Saykali and El-Sibai, 2014*) and recruits cytosolic proteins containing lipid-binding domains, such as the PH domain, to the plasma membrane (*Toker and Cantley, 1997*). ARL4C in pancreatic cancer cells preferred PIP3 to PIP2. Because PI3 kinase is one of the direct effector proteins of RAS (*Castellano and Downward, 2011*; *Rodriguez-Viciana et al., 1994*), RAS-dependent PI3 kinase activation and ARL4C expression could co-operatively function to promote pancreatic cancer invasion.

In conclusion, this study clarified that invasion of pancreatic cancer cells is promoted by ARL4C, of which expression is induced by KRAS and Wnt signaling, and that association of ARL4C with IQGAP1 and MMP14 at the tips of invasive pseudopods are essential for the invasive ability. The novel functions of ARL4C were confirmed by the mouse model. The inhibition of ARL4C expression by ARL4C ASO could directly inhibit invasive ability of pancreatic cancer cells and may indirectly affect the genes involved in invasion perhaps through the interaction between tumors and surrounding tissues. Because histological damage to the non-tumor regions was not observed after the administration of ARL4C ASO-1316 (*Harada et al., 2019*), ARL4C might represent an appropriate target for pancreatic cancer therapy.

# Materials and methods

## Key resources table

| Reagent type (species) or resource | Designation | Source or reference | Identifiers | Additional information |
|---|---|---|---|---|
| Strain, strain background (*Mus musculus*, male) | BALB/cAJcl-nu/nu | CLEA | | Ten-week-old |
| Cell line (*Homo sapiens*) | Lenti-X 293T | Takara Bio Inc | | |
| Cell line (*Homo sapiens*) | HeLaS3 | K.Matsumoto (Nagoya University, Aichi, Japan) | RRID:CVCL_0058 | |
| Cell line (*Homo sapiens*) | S2-CP8 | Cell Resource Center for Biomedical Research, Institute of Development, Aging and Cancer, Tohoku University | RRID:CVCL_F971 | |
| Cell line (*Homo sapiens*) | PANC-1 | RIKEN Bioresource Center Cell Bank | RRID:CVCL_0480 | |
| Cell line (*Homo sapiens*) | BxPC-3 | American Type Culture Collection | RRID:CVCL_0186 | |
| Cell line (*Homo sapiens*) | HPAF-II | American Type Culture Collection | RRID:CVCL_0313 | |
| Transfected construct (*Homo sapiens*) | ARL4C-EGFP | This paper | | Various mutants of ARL4C |
| Transfected construct (*Homo sapiens*) | ARL4C$^{G2A}$-EGFP | This paper | | Various mutants of ARL4C |
| Transfected construct (*Homo sapiens*) | ARL4C$^{T27N}$-EGFP | This paper | | Various mutants of ARL4C |
| Transfected construct (*Homo sapiens*) | ARL4C$^{Q72L}$-EGFP | This paper | | Various mutants of ARL4C |
| Transfected construct (*Homo sapiens*) | ARL4C$^{\Delta PBR}$-EGFP | This paper | | Various mutants of ARL4C |
| Transfected construct (*Homo sapiens*) | ARL4C$^{G2A}$-EGFP-Cterm | This paper | | Various mutants of ARL4C |
| Transfected construct (*Homo sapiens*) | ARL4C$^{\Delta PBR}$-EGFP-Cterm | This paper | | Various mutants of ARL4C |
| Transfected construct (*Homo sapiens*) | ARL4C$^{G2A}$-EGFP-GRP1PH | This paper | | Various mutants of ARL4C |
| Transfected construct (*Homo sapiens*) | ARL4C$^{G2A}$-EGFP-PLCδPH | This paper | | Various mutants of ARL4C |
| Transfected construct (*Homo sapiens*) | ARL4C-mCherry | This paper | | Various mutants of ARL4C |
| Transfected construct (*Homo sapiens*) | ARL4C-tdTomato | This paper | | Various mutants of ARL4C |

*Continued on next page*

*Continued*

| Reagent type (species) or resource | Designation | Source or reference | Identifiers | Additional information |
|---|---|---|---|---|
| Transfected construct (*Homo sapiens*) | ARL4C-FLAG-HA | This paper | | Various mutants of ARL4C |
| Biological sample (*Homo sapiens*) | Resected specimens of 57 patients with PDAC | Osaka University | | |
| Antibody | ARL4C (rabbit polyclonal) | Atlas Antibodies | #HPA028927 | (WB 1:1000, IHC 1:50) |
| Antibody | Clathrin (mouse monoclonal) | BD Biosciences | #610,500 | (WB 1:1000) |
| Antibody | EGR1 (rabbit monoclonal) | Cell Signaling Technology | #4,153 S | (WB 1:1000) |
| Antibody | β-catenin (mouse monoclonal) | BD Biosciences | #610,154 | (WB 1:1000) |
| Antibody | Ras (G12D) (rabbit monoclonal) | Cell Signaling Technology | #14,429 S | (WB 1:1000) |
| Antibody | Hsp90 (mouse monoclonal) | BD Biosciences | #610,419 | (WB 1:1000) |
| Antibody | HA (mouse monoclonal) | BioLegend | #901,502 | (WB 1:1000) |
| Antibody | HA (rat monoclonal) | Roche | #1867423001 | (ICC 1:100) |
| Antibody | GFP (rabbit polyclonal) | Life Technologies/Thermo Fisher Scientific | #A6455 | (WB 1:4000) |
| Antibody | GFP (mouse monoclonal) | Santa Cruz Santa Cruz Biotechnology | #sc-9996 | (WB 1:1000) |
| Antibody | FLAG (mouse monoclonal) | WAKO | #014–22,383 | (WB 1:1000) |
| Antibody | IQGAP1 (mouse monoclonal) | Santa Cruz Santa Cruz Biotechnology | #sc-376021 | (WB 1:1000, IHC 1:800, ICC 1:100) |
| Antibody | MMP14 (rabbit monoclonal) | Abcam | #ab51074 | (WB 1:1000, IHC 1:200, ICC 1:100) |
| Antibody | Cytohesin2 (rabbit polyclonal) | Proteintech Group, Inc | #67185–1-Ig | (ICC 1:100) |
| Antibody | Rac1 (mouse monoclonal) | BD Biosciences | #610,651 | (WB 1:1000) |
| Antibody | Cdc42 (rabbit polyclonal) | Cell Signaling Technology | #2,466 S | (WB 1:1000) |
| Antibody | CK19 (rabbit monoclonal) | Abcam | #ab52625 | (IHC 1:100) |
| Antibody | Mitochondria (mouse monoclonal) | Merck Millipore | #MAB1273 | (IHC 1:100) |
| Antibody | LYVE-1 (rabbit polyclonal) | Abcam | #ab14917 | (IHC 1:100) |
| Antibody | YAP/TAZ (rabbit monoclonal) | Cell Signaling Technology | #8,418 S | (ICC 1:100) |
| Antibody | Paxillin (mouse monoclonal) | BD Biosciences | #610,052 | (ICC 1:100) |

*Continued*

| Reagent type (species) or resource | Designation | Source or reference | Identifiers | Additional information |
|---|---|---|---|---|
| Antibody | FAK (mouse monoclonal) | BD Biosciences | #610,087 | (ICC 1:100) |
| Antibody | P-Paxillin (Y118) (rabbit polyclonal) | Cell Signaling Technology | #2,541 S | (ICC 1:100) |
| Antibody | P-FAK (Y397) (rabbit monoclonal) | Life Technologies/Thermo Fisher Scientific | #44,625 G | (ICC 1:100) |
| Antibody | Cortactin (mouse monoclonal) | Merck Millipore | #05–180 | (ICC 1:100) |
| Antibody | ARL4C (rabbit polyclonal) | This paper | SAJ5550275 | (ICC 1:100) |
| Recombinant DNA reagent | pEGFPC2-IQGAP1 (plasmid) | K.Kaibuchi (Nagoya University, Japan) | | |
| Recombinant DNA reagent | pEGFP-mCyth2 (plasmid) | J.Yamauchi (Tokyo University of Pharmacy and Life Science, Japan), | | |
| Recombinant DNA reagent | pAcGFP-mPlcd1PH (plasmid) | M.Matsuda (Kyoto University, Kyoto, Japan) | | |
| Recombinant DNA reagent | CSII-CMV-MCS-IRES2-Bsd (plasmid) | H.Miyoshi (RIKEN Bioresource Center, Ibaraki, Japan) | | |
| Recombinant DNA reagent | pEGFPN3-hARL4C (plasmid) | A.Kikuchi (Osaka University, Osaka, Japan) | | |
| Recombinant DNA reagent | pEGFPN3-hGRP1 (plasmid) | This paper | | Full length cDNAs of GRP1 ORF were reversely transcribed from mRNA extracted from MCF-7 cells. |
| Recombinant DNA reagent | pEGFPN3-hMMP14 (plasmid) | This paper | | Full length cDNAs of MMP14 ORF were reversely transcribed from mRNA extracted from U2OS cells. |
| Recombinant DNA reagent | mRFP-FKBP-5-ptase-dom | Addgene | 67,516 | |
| Recombinant DNA reagent | PM-FRB-CFP | Addgene | 67,517 | |
| Sequence-based reagent | siRNA: randomized control | This paper | | 5'-CAGTCGCGTTTGCGACTGG-3' |
| Sequence-based reagent | siRNA: human *IQGAP1#1* | This paper | | 5'-GCTGCACATAGTTGCCTTT-3' |
| Sequence-based reagent | siRNA: human *IQGAP1#2* | This paper | | 5'-CCCTAATGTAGAATGTCAT-3' |
| Sequence-based reagent | siRNA: human *CYTH2#1* | This paper | | 5'-GGATGGAGCTGGAGAACAT-3' |
| Sequence-based reagent | siRNA: human *CYTH2#2* | This paper | | 5'-GCAGTTTCTATGGAGCTTT-3' |
| Sequence-based reagent | siRNA: human *ARPC2#1* | This paper | | 5'-GCCTATATTCACACACGTA-3' |
| Sequence-based reagent | siRNA: human *ARPC2#2* | This paper | | 5'-CCTATATTCACACACGTAT-3' |
| Sequence-based reagent | siRNA: human *MMP14#1* | This paper | | 5'-GCAGCCTCTCACTACTCTT-3' |
| Sequence-based reagent | siRNA: human *MMP14#2* | This paper | | 5'-CCGACATCATGATCTTCTT-3' |

*Continued*

| Reagent type (species) or resource | Designation | Source or reference | Identifiers | Additional information |
|---|---|---|---|---|
| Sequence-based reagent | siRNA: human *KRAS#1* | This paper | | 5'-GCATCATGTCCTATAGTTT-3' |
| Sequence-based reagent | siRNA: human *KRAS#2* | This paper | | 5'-GTTGGAGCTGATGGCGTAG-3' |
| Sequence-based reagent | siRNA: human *CTNNB1#1* | This paper | | 5'-CCCACTAATGTCCAGCGTT-3' |
| Sequence-based reagent | siRNA: human *CTNNB1#2* | This paper | | 5'-GCATAACCTTTCCCATCAT-3' |
| Sequence-based reagent | Antisense oligonucleotide: randomized control | This paper | | T(Y)^a^g^A(Y)^g^a^G(Y)^t^a^5(Y)^c^c^A(Y)^t^c (Lower case = DNA; N(Y) = AmNA; 5(Y) = AmNA_mC; ^ = Phosphorothioated) |
| Sequence-based reagent | Antisense oligonucleotide: ARL4C-1316 | This paper | | G(Y)^5(Y)^A(Y)^t^a^c^c^t^c^a^g^g^T(Y)^A(Y)^a (Lower case = DNA; N(Y) = AmNA; 5(Y) = AmNA_mC; ^ = Phosphorothioated) |
| Sequence-based reagent | human *GAPDH_F* | This paper | PCR primers | 5'-TCCTGCACCACCAACTGCTT-3' |
| Sequence-based reagent | human *GAPDH_R* | This paper | PCR primers | 5'-TGGCAGTGATGGCATGGAC-3' |
| Sequence-based reagent | human *B2M_F* | This paper | PCR primers | 5'-TGCTGTCTCCATGTTTGATGTATC-3 |
| Sequence-based reagent | human B2M_R | This paper | PCR primers | 5'-TCTCTGCTCCCCACCTCTAAG-3' |
| Sequence-based reagent | human *ARL4C_F* | This paper | PCR primers | 5'-AGGGGCTGTGAAGCTGAGTA-3′ |
| Sequence-based reagent | human *ARL4C_R* | This paper | PCR primers | 5'-TTCCAGGCTGAAAAGCAGTT –3' |
| Sequence-based reagent | human *ARPC2_F* | This paper | PCR primers | 5'-AGATTTCGATGGGGTCCTCT-3' |
| Sequence-based reagent | human *ARPC2_R* | This paper | PCR primers | 5'-CCGGAAGATTTTCAAGGTCA-3' |
| peptide, recombinant protein | FLAG peptide | Sigma-Aldrich | F3290 | |
| Commercial assay or kit | Lipofectamine2000 transfection reagent | Life Technologies/Thermo Fisher Scientific | 11668019 | |
| Commercial assay or kit | Lipofectamine LTX reagent | Life Technologies/Thermo Fisher Scientific | 15338100 | |
| Commercial assay or kit | RNAiMAX | Life Technologies/Thermo Fisher Scientific | 13778075 | |
| Commercial assay or kit | ViaFect | Promega Corp. | E4981 | |
| Commercial assay or kit | TrypLE Express Enzyme | Thermo Fisher Scientific | 12604013 | |
| Commercial assay or kit | PrimeSTAR Max DNA Polymerase | Takara Bio Inc | R045A | |
| Commercial assay or kit | In-Fusion HD Cloning Kit | Clontech | 639,649 | |

*Continued on next page*

*Continued*

| Reagent type (species) or resource | Designation | Source or reference | Identifiers | Additional information |
|---|---|---|---|---|
| Commercial assay or kit | DakoReal EnVision Detection System | Dako | K500711-2 | |
| Commercial assay or kit | Peroxidase-Blocking Solution | Dako | S202386-2 | |
| Commercial assay or kit | G-Block | GenoStaff | GB-01 | |
| Commercial assay or kit | Blocking One Histo | nacalai tesque | 06349–64 | |
| Commercial assay or kit | rat tail type I collagen | Corning Inc | 354,236 | |
| Commercial assay or kit | DQ-collagen type I | Invitrogen | D12060 | |
| Commercial assay or kit | 3D microfluidic cell culture chip | AIM Biotech | DAX-1 | |
| Commercial assay or kit | QCM Gelatin Invadopodia Assay (Red) | Merck Millipore | ECM671 | |
| Commercial assay or kit | poly-D-lysine | Sigma-Aldrich | P6407 | |
| Commercial assay or kit | Matrigel Growth Factor Reduced | Corning Inc | 354,230 | |
| Commercial assay or kit | 6.5 mm Transwell with 8.0 µm Pore Polycarbonate Membrane Insert | Corning Inc | 3,422 | |
| Commercial assay or kit | BioCoat Matrigel Invasion Chambers with 8.0 µm PET Membrane | Corning Inc | 354,480 | |
| Commercial assay or kit | Annexin V-FITC Apoptosis Detection Kit | nacalai tesque | 15342–54 | |
| Commercial assay or kit | protein A Sepharose beads | GE Healthcare | 17078001 | |
| Commercial assay or kit | Dynabeads Protein G | Thermo Fisher Scientific | DB10003 | |
| Commercial assay or kit | Pierce Silver Stain for Mass Spectrometry | Thermo Fisher Scientific | 24,600 | |
| Commercial assay or kit | O.C.T. Compound | Sakura Finetek | | |
| Commercial assay or kit | NucleoSpin RNA | MACHEREY-NAGEL GmbH & Co. KG | 740,955 | |
| Commercial assay or kit | ReverTra Ace qPCR RT Master Mix | TOYOBO | FSQ-201 | |
| Commercial assay or kit | Pierce BCA Protein Assay Kit | Thermo Fisher Scientific | 23,227 | |
| Chemical compound, drug | PD184161 | Sigma-Aldrich | PZ0112 | |

*Continued on next page*

*Continued*

| Reagent type (species) or resource | Designation | Source or reference | Identifiers | Additional information |
|---|---|---|---|---|
| Chemical compound, drug | U0126 | Promega Corp. | V1121 | |
| Chemical compound, drug | Rapamycin | Cell Signaling Technology | 9,904 | |
| Chemical compound, drug | LY294002 | Cell Signaling Technology | 9,901 | |
| Chemical compound, drug | VivoGlo luciferin | Promega Corp. | P1043 | |
| Software, algorithm | HALO | Indica Labs | | RRID:SCR_018350 |
| Software, algorithm | NanoZoomer-SQ | Hamamatsu Photonics K.K. | | |
| Software, algorithm | UCSC Xena browser | http://xena.ucsc.edu | | RRID:SCR_018938 |
| Software, algorithm | Kaplan–Meier plotter | http://www.kmplot.com | | RID:SCR_018753 |
| Software, algorithm | GraphPad Prism 8 | GraphPad Software. | | RRID:SCR_002798 |
| Software, algorithm | Excel Toukei | ESUMI Co., Ltd. | | |
| Software, algorithm | Imaris | Bitplane | | RRID:SCR_007370 |
| Software, algorithm | Image J | National Institutes of Health | | RRID:SCR_003070 |
| Software, algorithm | Living Image 4.3.1 Software | Caliper Life Sciences | | RRID:SCR_014247 |
| Software, algorithm | ikra v1.2.2 | https://zenodo.org/record/3606888 (*Yu et al., 2019*) | | |
| Software, algorithm | iDEP.90 | http://bioinformatics.sdstate.edu/idep90/ (*Ge et al., 2020*) | | |
| Software, algorithm | Ingenuity Pathway Analysis | IPA; Qiagen | | RRID:SCR_008653 |
| Other | LSM880 laser scanning microscope | Carl Zeiss | | |
| Other | BZ-9000 | Keyence | | |
| Other | IVIS imaging system | Xenogen Corp. | | |
| Other | Hoechst33342 | Invitrogen | H1399 | |
| Other | Alexa Fluor 488 Phalloidin | Invitrogen | A12379 | |
| Other | Alexa Fluor 546 Phalloidin | Invitrogen | A22283 | |
| Other | Alexa Fluor 647 Phalloidin | Invitrogen | A22287 | |

## Materials and chemicals

HeLaS3 cells were kindly provided by Dr. K. Matsumoto (Nagoya University, Aichi, Japan) in May 2002. S2-CP8 pancreatic cancer cells were purchased from Cell Resource Center for Biomedical Research, Institute of Development, Aging and Cancer, Tohoku University, in April 2014. Lenti-X 293T (X293T)

cells were purchased from Takara Bio Inc (Shiga, Japan) in October 2011. PANC-1 cells were purchased from RIKEN Bioresource Center Cell Bank (RIKEN BRC, Tsukuba, Japan) in October 2014. BxPC-3 cells were purchased from American Type Culture Collection (ATCC, Manassas, VA, USA) in May 2018. HPAF-II cells were purchased from ATCC in July 2017. S2-CP8, X293T, HeLaS3, and HPAF-II cells were grown in Dulbecco's modified Eagle's medium (DMEM) supplemented with 10 % fetal bovine serum (FBS). PANC-1 and BxPC-3 cells were grown in RPMI-1640 supplemented with 10 % FBS. All cell lines were authenticated using short tandem repeat profiling by BEX CO., LTD (Tokyo, Japan) and tested negative for Mycoplasma using e-Myco Mycoplasma PCR Detection Kit (iNtRON Biotechnology, Inc, Gyeonggi-do, Korea).

S2-CP8 cells stably expressing GFP, ARL4C-EGFP, ARL4C$^{G2A}$-EGFP, ARL4C$^{T27N}$-EGFP, ARL4C$^{Q72L}$-EGFP, ARL4C$^{\Delta PBR}$-EGFP, ARL4C$^{G2A}$-EGFP-Cterm, ARL4C$^{\Delta PBR}$-EGFP-Cterm, ARL4C$^{G2A}$-EGFP-GRP1$^{PH}$, ARL4C$^{G2A}$-EGFP-PLCδ$^{PH}$, ARL4C-mCherry, ARL4C-tdTomato, EGFP-IQGAP1, and luciferase were generated using lentivirus as described previously (*Kimura et al., 2016*). BxPC-3 cells stably expressing EGFP or ARL4C-EGFP were generated using lentivirus. Lentiviral vector CSII-CMV-MCS-IRES2-Bsd harboring a cDNA was transfected with the packaging vectors pCAG-HIV-gp and pCMV-VSV-G-RSV-Rev into X293T cells using Lipofectamine2000 transfection reagent (Life Technologies/Thermo Fisher Scientific, Carlsbad, CA, USA). To generate S2-CP8 stable cells above, $1 \times 10^5$ parental cells/well in a 12-well plate were treated with lentiviruses and 5 μg/mL polybrene, centrifuged at 1200 x *g* for 30 min, and incubated for 24 h. The cells were selected and maintained in the medium containing 10 μg/mL Blasticidin S.

ARL4C or IQGAP1 knockout cells were generated as previously described (*Fujii et al., 2016*). The target sequences for human ARL4C, 5'-CTTCTCGGTGTTGAAGCCGA-3', and human IQGAP1, 5'-CACCGTGGGGTCTACCTTGCCAAAC-3' were designed with the help of the CRISPR Genome Engineering Resources (http://www.genome-engineering.org/crispr/). The plasmids expressing hCas9 and single-guide RNA (sgRNA) were prepared by ligating oligonucleotides into the BbsI site of pX330 (addgene #42230). The plasmid pX330 with sgRNA sequences targeting ARL4C, IQGAP1 and Blasticidin resistance was introduced into S2-CP8 cells using Lipofectamine LTX reagent (Life Technologies/Thermo Fisher Scientific) according to manufacturer's instructions and the transfected cells were selected in medium containing 5 μg/mL Blasticidin S for 2 days. Single colonies were picked, mechanically disaggregated, and replated into individual wells of 24-well plates.

ARL4C ASO-1316 and 6-carboxyfluorescein (FAM)-labeled ARL4C ASO-1316 were synthesized by GeneDesign (Osaka, Japan) as described (*Harada et al., 2019*). The sequences of the ASOs are listed in *Supplementary file 1 table 5*. S2-CP8 cell were transfected with ASOs at 10 nmol/L using RNAiMAX (Life Technologies/Thermo Fisher Scientific) in antibiotics-free medium. The transfected cells were then used for experiments conducted at 48 hr after transfection.

Anti-ARL4C polyclonal antibody (SAJ5550275) for immunoprecipitation and immunocytochemistry was generated in rabbits by immunization with recombinant human ARL4C. Antibodies used in this study are shown in *Supplementary file 1 table 6*.

The following drugs were used: PD184161 (Sigma-Aldrich Co, St. Louis, MO, USA); U0126 (Promega Corp., Madison, WI, USA); Rapamycin (Cell Signaling Technology, Beverly, MA, USA); LY294002 (Cell Signaling Technology); and VivoGlo luciferin (Promega Corp.).

Plasmid construction pEGFPC2-IQGAP1, pEGFP-mCyth2, pAcGFP-mPlcd1$^{PH}$, and CSII-CMV-MCS-IRES2-Bsd were kindly provided by K. Kaibuchi (Nagoya University, Japan), J. Yamauchi (Tokyo University of Pharmacy and Life Science, Japan), M. Matsuda (Kyoto University, Kyoto, Japan), and H. Miyoshi (RIKEN Bioresource Center, Ibaraki, Japan), respectively.

To generate plasmid DNA with mutated codons or deletions, site-directed mutagenesis method was performed using PrimeSTAR Max DNA Polymerase (Takara Bio Inc, Shiga, Japan). To generate plasmid DNA with insertions, PCR amplified fragments and linearized vector by restriction enzyme digestion were assembled using In-Fusion HD Cloning Kit (Takara Bio Inc).

pEGFPN3-ARL4C was constructed as previously described (*Matsumoto et al., 2014*). Full length cDNAs of GRP1 and MMP14 ORF were reversely transcribed from mRNA extracted from MCF-7 cells and U2OS cells, respectively. Linear double strand oligonucleotides of the C-terminal 22 amino acids of KRAS, which includes the PBR and CAAX motifs, were synthesized, and the oligonucleotides were inserted into C terminal of ARL4C-EGFP or ARL4C-FLAG-HA using In-Fusion HD Cloning Kit (Takara Bio Inc).

Standard recombinant DNA techniques mentioned above were used to construct the following plasmids: pEGFPN3-ARL4C, pEGFPN3-ARL4C$^{G2A}$, pEGFPN3-ARL4C$^{T27N}$, pEGFPN3-ARL4C$^{Q72L}$, pEGFPN3-ARL4C$^{\Delta PBR}$, pEGFPN3-ARL4C$^{G2A}$-EGFP-PLC$\delta^{PH}$, pEGFPN3-ARL4C$^{G2A}$-EGFP-GRP1$^{PH}$, pEGFPN3-ARL4C$^{G2A}$-EGFP-Cterm, pEGFPN3-ARL4C$\Delta$PBR-EGFP-Cterm, pEGFPC1-CHD, pEGF-PC1-IQ, pEGFPC1-WW, pEGFPC1-IR, pEGFPC1-GRD, pEGFPC1-RGCT, pcDNA3-ARL4C-FLAG-HA, pcDNA3-ARL4C$^{G2A}$-FLAG-HA, pcDNA3-ARL4C$^{\Delta PBR}$-FLAG-HA, pcDNA3-ARL4C$^{G2A}$-FLAG-HA-Cterm, pcDNA3-ARL4C$^{\Delta PBR}$-FLAG-HA-Cterm, pcDNA3-FLAG-HA-IQGAP1, pmCherryN1-ARL4C, pmCherryN1-MMP14, pmCherryN1-MMP14$\Delta$C($\Delta$563–582), pCAG-ARL4C-tdTomato. To construct lentiviral vectors harboring EGFP, ARL4C-EGFP, ARL4C$^{G2A}$-EGFP, ARL4C$^{T27N}$-EGFP, ARL4C$^{Q72L}$-EGFP, ARL4C$^{\Delta PBR}$-EGFP, ARL4C$^{G2A}$-EGFP-Cterm, ARL4C$^{\Delta PBR}$-EGFP-Cterm, ARL4C$^{G2A}$-EGFP-PLC$\delta$PH, ARL4C$^{G2A}$-EGFP-GRP1PH, EGFP-IQGAP1, ARLC-mCherry, MMP14-mCherry, MMP14$\Delta$C-mCherry, ARL4C-tdTomato were cloned into CSII-CMV-MCS-IRES2-Bsd provided by Dr. H. Miyoshi (RIKEN Bioresource Center, Ibaraki, Japan).

## Patients and cancer tissues

The present study involved 57 presurgical untreated patients with PDAC and ages ranging from 47 to 87 years (median, 70 years) who underwent surgical resection at Osaka University between April 2001 and April 2015. Tumors were staged according to the Union for International Cancer Control (UICC) TNM staging system. Resected specimens were fixed in 10 % (vol/vol) formalin, processed for paraffin embedding, and were sectioned at 5 µm thickness and stained with hematoxylin and eosin (H&E) or immunoperoxidase for independent evaluations. The protocol for this study was approved by the ethical review board of the Graduate School of Medicine, Osaka University, Japan (No. 13455), under the Declaration of Helsinki, and written informed consent was obtained from all patients. The study was performed in accordance with Committee guidelines and regulations.

## Immunohistochemical studies

Immunohistochemical studies were performed as previously described (*Fujii et al., 2015*) with modification. Briefly, all tissue sections were stained using a DakoReal EnVision Detection System (Dako, Carpentaria, CA, USA) in accordance with the manufacturer's recommendations. Formalin-fixed, paraffin-embedded tissue specimens for examination were sectioned at 5 µm thickness. Heat-induced epitope retrieval was performed using Decloaking Chamber NxGen (Biocare Medical, Walnut Creek, CA, USA). Tissue peroxidase activity was blocked with Peroxidase-Blocking Solution (Dako) for 30 min, and the sections were then incubated with G-Block (GenoStaff, Tokyo, Japan) or Blocking One Histo (nacalai tesque, Kyoto, Japan) for 30 min or 10 min, respectively, to block nonspecific antibody binding sites. Tissue specimens were treated with anti-ARL4C (1:100), anti-IQGAP1 (1:800), or anti-MMP14 (1:100) antibody for 3 hr at room temperature. Then, the specimens were detected by incubating with goat anti-rabbit or anti-rabbit/mouse IgG-HRP for 1 h and subsequently with DAB (Dako). The tissue sections were then counterstained with 0.1 % (wt/vol) hematoxylin. ARL4C expression was considered high when the total area of the tumor stained with anti-ARL4C antibody exceeded 5 %. IQGAP1 expression was considered high when the total area of the tumor stained with anti-IQGAP1 antibody exceeded 40 %.

IQGAP1 staining positivity in PDAC patients was measured using HALO (Indica Labs, Corrales, NM, USA). The threshold for positive or negative staining was based on the optical density of the staining: regions above the positivity threshold were scored according to the optical density threshold set in the module; weakly positive is shown in yellow and strongly positive in red. The samples were viewed and analyzed using NanoZoomer-SQ (Hamamatsu Photonics K.K., Shizuoka, Japan).

## Clinical data analyses using open sources

The data on ARL4C and IQGAP1 mRNA expression in pancreatic adenocarcinoma were obtained from the UCSC Xena browser (http://xena.ucsc.edu). Tumors and normal samples in the UCSC Xena browser were derived from The Cancer Genome Atlas (TCGA) and Genotype-Tissue Expression (GTEx) projects. Differential analysis was performed using a two-tailed Student's t-test. The correlations of overall survival rates with ARL4C, IQGAP1, and MMP14 expression in pancreatic cancer in TCGA datasets were analyzed using a Kaplan–Meier plotter (http://www.kmplot.com) and visualized

using GraphPad Prism 8 (GraphPad Software. San Diego, CA, USA). High and low expression groups were classified by auto select best cutoff. p Values and *r* values were calculated using GraphPad Prism.

## 3D gel invasion assay using a 3D microfluidic cell culture chip

Collagen gels were made by diluting and neutralizing rat tail type I collagen (Corning Inc, Corning, NY, USA) in PBS and 12.1 mM NaOH, and were adjusted to 2 mg/mL. DQ-collagen type I (Life Technologies/Thermo Fisher Scientific, Carlsbad, CA, USA) was mixed with collagen gels at a final concentration of 25 µg/mL. The gel channel of 3D microfluidic cell culture chip (AIM Biotech, Biopolis Rd, Singapore) was filled with collagen solution and incubated at 37 °C for at least 1 hr to polymerize collagen. After hydration of medium channels, a cell suspension ($1 \times 10^4$ cells) in serum-free cell culture medium with 0.2 % BSA was injected into one of the ports at the medium channel. The opposite medium channel was filled with cell culture medium containing 10 % FBS to create a chemoattractant gradient across the collagen gel. The cells were then incubated for 3 days and fixed for 15 min at room temperature in PBS containing 4 % (w/v) paraformaldehyde. Then, the cells were permeabilized and blocked in PBS containing 0.5 % (w/v) Triton X-100 and 40 mg/mL BSA for 30 min and stained with the indicated antibodies. The samples were viewed and analyzed under an LSM880 laser scanning microscope (Carl Zeiss, Jana, Germany). Reconstruction of confocal z-stack images into 3D animations and analysis of 4D images were performed using Imaris (Bitplane, Belfast, UK).

## Invadopodia assay

QCM Gelatin Invadopodia Assay (Red) (Merck Millipore, Burlington, MA, USA) was used in accordance with the manufacturer's protocol. Briefly, poly-L-lysine–coated coverslips were treated with glutaraldehyde. The coverslips were then incubated with Cy3-labeled gelatin, followed by culture medium quenching of free aldehydes. Cells ($6 \times 10^4$ cells) were seeded onto the gelatin-coated coverslips and incubated for 4 hr. After incubation, the cells were fixed for 20 min at room temperature in phosphate-buffered saline (PBS) containing 4 % (w/v) paraformaldehyde and permeabilized in PBS containing 0.2 % (w/v) Triton X-100 for 10 min. After being blocked in PBS containing 0.2 % (w/v) BSA for 30 min, the cells were immunohistochemically stained. The samples were viewed and analyzed under an LSM880 laser scanning microscope (Carl Zeiss, Jana, Germany).

## 2D culture on poly-D-lysine– or matrigel-coated dishes

Cells grown on glass coverslips coated with poly-D-lysine (Sigma-Aldrich) or Matrigel Growth Factor Reduced (Corning) were fixed for 10 min at room temperature in PBS containing 4 % (w/v) paraformaldehyde and permeabilized in PBS containing 0.1 % (w/v) saponin (Sigma-Aldrich) or 0.2 % (w/v) Triton X-100 for 10 min. The cells were then blocked in PBS containing 0.2 % (w/v) BSA for 30 min. They were then incubated with primary antibodies for 3 hr at room temperature and with secondary antibodies in accordance with the manufacturer's protocol (Life Technologies/Thermo Fisher Scientific). For cell surface MMP14 staining, samples were incubated with anti-MMP14 antibody for 3 hr at room temperature without permeabilization. The samples were viewed and analyzed under an LSM880 laser scanning microscope (Carl Zeiss).

## Migration and invasion assays

Migration and invasion assays were performed using a modified Boyden chamber (6.5 mm Transwell with 8.0 µm Pore Polycarbonate Membrane Insert; Corning) and a Matrigel-coated modified Boyden chamber (BioCoat Matrigel Invasion Chambers with 8.0 µm PET Membrane; Corning), respectively as described previously (*Kurayoshi et al., 2006*; *Matsumoto et al., 2014*). In the standard conditions, S2-CP8 cells ($2.5 \times 10^4$ cells) were seeded in the upper side of Boyden Chamber. In GFP-expressing S2-CP8 cells, after 4 h (migration assay) or 24 hr (invasion assay, except for *Figure 6E*) incubation with control ASO, 122 cells (average) and 126 cells (average), respectively, were observed in the lower side chamber in the one field of view under fluorescence microscope (BZ-9000, Keyence, Osaka, Japan) using a 10 x air objective. In *Figure 6E*, cells were observed after 20 hr incubation with ASO. Migration and invasion rates of cells expressing ARL4C, IQGAP1, and MMP14 mutants were calculated as the percentages of the same cells transfected with control ASO or siRNA.

## 3D type I collagen gel culture

Collagen gels were made by diluting and neutralizing rat tail type I collagen (Corning) in PBS and 12.1 mM NaOH, and were adjusted to 2 mg/mL. Then, 140 µL of cell-embedded collagen gels (1

$\times 10^6$ cells/mL) were overlaid onto glass coverslips in a 24-well plate and allowed to polymerize for at least 1 hr at 37 °C and 5 % $CO_2$. After polymerization, growth medium was added on top of the collagen gel. The cells were then incubated for 3 days and fixed for 15 min at room temperature in PBS containing 4 % (w/v) paraformaldehyde. Then, the cells were permeabilized and blocked in PBS containing 0.5 % (w/v) Triton X-100 and 40 mg/mL BSA for 30 min and incubated with primary antibodies for 3 h at room temperature and secondary antibodies in accordance with the manufacturer's protocol (Life Technologies/Thermo Fisher Scientific). The samples were viewed and analyzed under an LSM880 laser scanning microscope using a 20 x air objective (Carl Zeiss). In the standard conditions (for *Figure 3L*) with BxPC-3/ARL4C-GFP cells treated with control ASO, the number of cells with pseudopods and the total number of cells were 15 (average) and 76 (average), respectively, in the one field of view under an LSM880 laser scanning microscope (Carl-Zeiss) using a 20 x air objective. The percentages of cells with pseudopods compared with the total number of cells in the presence of control siRNA or IQGAP1 siRNA were calculated.

Inducible recruitment of phospholipid phosphatases mRFP-FKBP-5-ptase-dom and PM-FRB-CFP plasmids were obtained from Addgene (deposited by the laboratory of T. Balla). S2-CP8 cells were then transiently transfected with both mRFP-FKBP-5-ptase-dom and PM-FRB-CFP (0.5 µg/well of a six-well plate for each vector) with ViaFect (Promega Corp.). After 24 hr culture, the cells were treated with 100 nM rapamycin or 50 µM LY294002 for 30 min before fixation.

### Cytotoxic assay

Cells transfected with control ASO or ARL4C ASO-1316 were cultured on Matrigel coated dish for 3.5 days, and dissociated using TrypLE Express (Thermo Fisher Scientific). Suspension of cells was stained with Hoechst 33342 or propidium iodide (PI) using Annexin V-FITC Apoptosis Detection Kit (nacalai tesque). The samples were viewed and analyzed under an LSM880 laser scanning microscope (Carl Zeiss), and the number of PI-positive cells was divided by the total number of nuclei stained with Hoechst 33342.

### Isolation of ARL4C-interacting protein

Confluent X293T cells transiently transfected with ARL4C-FLAG-HA in two 10 cm culture dishes were harvested and lysed in 800 µL of lysis buffer (25 mM Tris-HCl [pH7.5], 50 mM NaCl, 0.5 % TritonX-100) with protease inhibitors (nacalai tesque). After 10 min of centrifugation, lysates were incubated with 40 µL of 50 % slurry of anti-FLAG Affinity Gel (Sigma-Aldrich) for 30 min, and then add another 40 µL and incubated for 30 min. Beads were washed three times with 1 mL of lysis buffer. Recovered beads were incubated once with FLAG peptide (0.5 mg/mL) to elute proteins in 80 µL of PBS for 30 min at 4 °C. Then, the supernatant was precleaned with 40 µL of 50 % slurry of protein A Sepharose beads (GE Healthcare, Chicago, IL, USA) for 30 min at 4 °C. The precleaned lysates were incubated with 2 µg of anti-HA antibody (Santa Cruz, Dallas, TX, USA) and 50 µL of 50 % slurry of protein A Sepharose beads for 1 hr at 4 °C. Beads were washed three times with 1 mL of lysis buffer, and bound complexes were dissolved in 50 µL of Laemmli's sample buffer. The ARL4C-FLAG-HA-interacting proteins were detected by Pierce Silver Stain for Mass Spectrometry (Life Technologies/Thermo Fisher Scientific). Six bands (arrowheads in *Figure 3A*) were cut from the gel and analyzed by mass spectrometry.

### Immunoprecipitation

Immunoprecipitation was performed as described previously with modification (*Matsumoto et al., 2014*). For *Figure 3C* S2-CP8 cells (60 mm diameter dish) were lysed in 300 µL of lysis buffer (25 mM Tris–HCl pH 7.5, 50 mM NaCl, 0.5 % Triton-X100) with protease inhibitors (nacalai tesque) for 10 min on ice. After centrifugation, the supernatant was collected and pre-cleaned using 30 µL of Dynabeads Protein G (Thermo Fisher Scientific). After pre-cleaning, lysates were rotated with complex of Dynabeads (50 µL) and antibody (3.6 µg) for 10 min at room temperature. The beads were then washed with lysis buffer three times, and finally suspended in Laemmli's sample buffer.

### The RAC1 activity assay

The RAC1 activity assay was performed as described (*Matsumoto et al., 2014*). Briefly, cells were lysed in 400 µL of RAC1 assay buffer (20 mM Tris–HCl [pH 7.5], 150 mM NaCl, 1 mM dithiothreitol, 10 mM MgCl2, 1 % Triton-X100) with protease inhibitors (nacalai tesque) containing 20 µg of

glutathione-S-transferase (GST)-CRIB. After the lysates were centrifuged at 20,000 *g* for 10 min, the supernatants were incubated with glutathione-Sepharose (20 µl each) for 2 h at 4 °C. The beads were then washed with RAC1 assay buffer three times, and finally suspended in Laemmli's sample buffer. The precipitates were probed with the anti-RAC1 antibody.

## Imaging of ASO accumulation in tumor-bearing mice

Orthotopic transplantation was performed as described previously (*Kim et al., 2009*). Ten days after the transplantation, 150 µg/animal (approximately 7.5 mg/kg) of 6-FAM-ARL4C ASO-1316 was subcutaneously administered. Four h after the injection, the fluorescence intensities of various organs were measured ex vivo using the IVIS imaging system (Xenogen Corp.). After ex vivo imaging, unfixed mouse pancreas tissues were frozen in an O.C.T. Compound (Sakura Finetek, Tokyo, Japan)/sucrose mixture [1:1 (v/v) OCT and 1 x PBS containing 30 % sucrose]. Freshly frozen tissues were sectioned at 10 µm and fixed for 30 min at room temperature in PBS containing 4 % (w/v) paraformaldehyde. The cells were then permeabilized and blocked in PBS containing 0.5 % (w/v) Triton X-100 and 40 mg/mL BSA for 30 min and stained with the indicated antibodies. The samples were viewed and analyzed under an LSM880 laser scanning microscope (Carl Zeiss).

## Orthotopic xenograft tumor assay

An orthotopic transplantation assay was performed as described previously (*Kim et al., 2009*) with modification. Ten-week-old male BALB/cAJcl-nu/nu mice (nude mice; CLEA, Tokyo, Japan) were anesthetized and received an orthotopic injection of S2-CP8 cells into the mid-body of the pancreas using a 27 G needle ($5 \times 10^5$ cells suspended in 100 µL of HBSS with 50 % Matrigel). ASOs (50 µg/mouse, approximately 2.5 mg/kg) were administered subcutaneously twice a week from day 3. To evaluate the knockdown efficiency of ARL4C ASO-1316, tumor tissues were harvested from tumor-bearing mice 8 days after transplantation. Total RNAs were isolated using NucleoSpin RNA (MACHEREY-NAGEL GmbH & Co. KG, Dueren, Germany), and complementary DNAs were synthesized using ReverTra Ace qPCR RT Master Mix (TOYOBO, Osaka, Japan). For extraction of tissue proteins, tumor samples were lysed in 150 µL of lysis buffer (20 mM Tris–HCl [pH 8.0], 137 mM NaCl, 10 % glycerol, 1% NP40) and homogenized using Biomasher II (KANTO CHEMICAL CO.,Inc, Tokyo, Japan). Debris was removed by centrifugation and finally suspended in Lammli's buffer. Protein concentration was determined with Pierce BCA Protein Assay Kit (Thermo Fisher Scientific). The band intensities of western blotting were calculated using Image J (National Institutes of Health, USA). To assess the effect of ARL4C ASO-1316 on tumor progression, tumor burden was measured once a week using the IVIS imaging system (Xenogen Corp., Alameda, CA, USA). For the in vivo imaging, 100 µL of VivoGlo luciferin (30 mg/mL) was intraperitoneally administered and the bioluminescence imaging was performed 8 min later. The region of interest (ROI) was selected and the radiance values were measured with Living Image 4.3.1 Software (Caliper Life Sciences, Hopkinton, MA, USA). The mice were euthanized 28 days after transplantation. Tumor weights and numbers of mesenteric lymph nodes (diameter of lymph nodes > 1 mm) were measured. All protocols used for the animal experiments in this study were approved by the Animal Research Committee of Osaka University, Japan (No. 26-032-048).

## RNA sequencing

Sequenced reads were preprocessed by Trim Galore! v0.6.3 and quantified by Salmon v0.14.0 with the flags gcBias and validateMappings. GENCODE vM21 annotation was used as the transcript reference. The quantified transcript-level scaled TPM was summarized into a gene-level scaled TPM by using the R package tximport v1.6.0. All procedures were implemented using the RNAseq pipeline ikra v1.2.2 [https://zenodo.org/record/3606888 (*Yu et al., 2019*)] with the default parameters. Downstream analysis was conducted with an integrative RNAseq analysis platform, iDEP.90. After normalization with VST, principal component analysis was conducted. Hierarchical clustering was performed on the top 1,000 genes in terms of their standard deviation. Finally, DEGs were selected with a log2 fold change >1 and false discovery rate <0.1.

## Ingenuity pathway analysis (IPA)

DEGs identified from RNA sequence data were subjected to Ingenuity Pathway Analysis (IPA; Qiagen, Hilden, Germany). This analysis examines DEGs that are known to affect each biological function and

compares their direction of change to what is expected from the literature. To infer the activation states of implicated biological functions, two statistical quantities, Z-score and p value, were used. A positive or negative Z-score value indicates that biological functions are predicted to be activated or inhibited in the ARL4C ASO-1316–treated group relative to the control ASO-treated group. A negative Z-score means that the indicated biological functions are inhibited by ARL4C ASO-1316. The p value, calculated with the Fisher's exact test, reflects the enrichment of the DEGs on each pathway. For stringent analysis, only biological functions with a |Z-score| > 2 were considered significant.

## Statistics and reproducibility

Biological replicates are replicates on independent biological samples versus technical replicates that use the same starting samples. All experiments in this study were repeated using biological replicates. A minimum of three biological replicates were analyzed for all samples, and the results are presented as the mean ± s.d. or s.e.m. The cumulative probabilities of overall survival were determined using the Kaplan–Meier method; a log-rank test was used to assess statistical significance. The Student's t-test or Mann–Whitney test was used to determine if there was a significant difference between the means of two groups. One-way analysis of variance (ANOVA) with Bonferroni tests was used to compare three or more group means. Statistical analysis was performed using Excel Toukei (ESUMI Co., Ltd., Tokyo, Japan) and GraphPad Prism 8 (GraphPad Software, La Jolla, CA, USA); p < 0.05 was considered statistically significant. In box and whiskers plots, the top and bottom horizontal lines represent the 75th and the 25th percentiles, respectively, and the middle horizontal line represents the median. The size of the box represents the interquartile range and the top and bottom whiskers represent the maximum and the minimum values, respectively.

## Others

The siRNAs and primers used in these experiments are listed in *Supplementary file 1 tables 7 and 8*, respectively. 2.5D Matrigel growth assay and quantitative PCR were performed as described previously (*Matsumoto et al., 2019*; *Sato et al., 2010*).

# Acknowledgements

We thank the NGS core facility of the Genome Information Research Center at the Research Institute for Microbial Diseases of Osaka University for the data analysis support.

This study was supported by Eiji Oiki, Yuri Terao, and Center for Medical Research and Education, Graduate School of Medicine, Osaka University. Also this study was supported by Saki Ishino and Center of Medical Innovation and Translational Research, Osaka University.

# Additional information

### Competing interests

Kensaku Shojima: The other authors declare that no competing interests exist.

### Funding

| Funder | Grant reference number | Author |
| --- | --- | --- |
| Ministry of Education, Culture, Sports, Science and Technology | 16H06374 | Akira Kikuchi |
| Ministry of Education, Culture, Sports, Science and Technology | 18H04861 | Akira Kikuchi |
| Ministry of Education, Culture, Sports, Science and Technology | 18H05101 | Akira Kikuchi |

| Funder | Grant reference number | Author |
|---|---|---|
| Yasuda Memorial Medical Foundation | | Akira Kikuchi |
| Ichiro Kanehara Foundation for the Promotion of Medical Sciences and Medical Care | | Akira Kikuchi |
| Osaka University | | Akira Kikuchi |

The funders had no role in study design, data collection and interpretation, or the decision to submit the work for publication.

## Author contributions
Akikazu Harada, Conceptualization, Data curation, Formal analysis, Investigation, Methodology, Project administration, Visualization, Writing – original draft, Writing – review and editing; Shinji Matsumoto, Conceptualization, Investigation, Methodology, Project administration, Writing – original draft, Writing – review and editing; Yoshiaki Yasumizu, Investigation, Writing – review and editing; Kensaku Shojima, Investigation, Methodology; Toshiyuki Akama, Methodology; Hidetoshi Eguchi, Resources, Writing – review and editing; Akira Kikuchi, Conceptualization, Funding acquisition, Project administration, Supervision, Writing – original draft, Writing – review and editing

## Author ORCIDs
Akikazu Harada http://orcid.org/0000-0002-8123-652X
Shinji Matsumoto http://orcid.org/0000-0003-1804-264X
Yoshiaki Yasumizu http://orcid.org/0000-0002-9872-4909
Akira Kikuchi http://orcid.org/0000-0003-3378-9522

## Ethics
All protocols used for the animal experiments in this study were approved by the Animal Research Committee of Osaka University, Japan (No. 26-032-048). This information is mentioned in the 'Materials and Methods' section.

## Decision letter and Author response
Decision letter https://doi.org/10.7554/eLife.66721.sa1
Author response https://doi.org/10.7554/eLife.66721.sa2

# Additional files

## Supplementary files
• Supplementary file 1. Supplementary information for the data and methods supporting the article.
• Transparent reporting form
• Source data 1. Supplementary File 1 Table 1.
• Source data 2. Supplementary File 1 Table 2.

## Data availability
-Sequencing data have been deposited in DDBJ under accession codes DRA011537. -All data generated or analysed during this study are included in the manuscript and supporting files. Source data files have been provided for Figures 1-7, Figure 2-figure supplement 1, Figure 2-figure supplement 2, Figure 2-figure supplement 3, Figure 3-figure supplement 1, Figure 3-figure supplement 2, Figure 4-figure supplement 1, Figure 6-figure supplement 1, Figure 7-figure supplement 1, Supplementary File 1 Table 1, and Supplementary File 1 Table 2.

The following dataset was generated:

| Author(s) | Year | Dataset title | Dataset URL | Database and Identifier |
|---|---|---|---|---|
| Akikazu H | 2021 | Effects of ARL4C ASO on an orthotopic transplantation model | https://ddbj.nig.ac.jp/ DRASearch/ | DRASearch, DRA011537 |

The following previously published datasets were used:

| Author(s) | Year | Dataset title | Dataset URL | Database and Identifier |
|---|---|---|---|---|
| The Cancer Genome Atlas (TCGA) Research Network | 2020 | A combined cohort of TCGA, TARGET and GTEx samples | https://xenabrowser. net/datapages/? cohort=TCGA% 20TARGET%20GTEx& addHub=https% 3A%2F%2Fxena. treehouse.gi.ucsc. edu&removeHub= https%3A%2F% 2Fpcawg.xenahubs. net | UCSC Xena, TCGA TARGET GTEx |

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
