## [Decision Letter]

**Acceptance summary:**

This work reveals a hitherto unknown mechanism by which KRAS and ARL4C regulate the invasion of pancreatic cancer cells. These results may reveal new targets for the prevention of pancreatic cancer metastasis.

**Decision letter after peer review:**

Thank you for submitting your article "Recruitment of KRAS downstream target ARL4C to membrane protrusions accelerates pancreatic cancer cell invasion." for consideration by *eLife*. Your article has been reviewed by 3 peer reviewers, one of whom is a member of our Board of Reviewing Editors, and the evaluation has been overseen by Erica Golemis as the Senior Editor. The following individuals involved in review of your submission have agreed to reveal their identity: Ivan Robert Nabi (Reviewer #2); Hon Leong (Reviewer #3).

The current manuscript describes a functional relationship between ARLC4, IQGAP1, and MMP14 in pancreatic cancer cells, such that ARLC4 is up-regulated in response to MAPK and Wnt signaling and colocalizes with IQGAP1 and MMP14 at membrane protrusions. These changes in expression and localization were associated with an increase in cellular invasion. Finally, the authors demonstrated that systemically delivered antisense oligonucleotides targeting ARL4C may prevent pancreatic cancer metastasis. Overall, this study reveals a hitherto undescribed mechanism for the regulation of pancreatic cancer cell invasion. Many of the in vitro cell biology experiments linking ARL4C to invasion are thorough; however, a more extensive and focused analysis of membrane protrusion localization is still required. Moreover, ties to clinical correlates as well as the in vivo metastasis assays could be improved upon.

Many strengths were noted:

Reviewers commended the comprehensive analysis of ARL4C in migration and 3D gel invasion, matrix degradation, and expression in tumors including mutational analyses that defined ARL4C domains that mediate these interactions. The potential to target ARL4C with ASO was deemed to be interesting and novel. The IHC data strongly supports expression of ARL4C in perineural lesions, which is a strong risk factor for poor OS and HR in pancreatic cancer.

Essential revisions:

A number of issues were raised that need to be addressed with experimentation, prior to publication. Specifically, studies that more concretely demonstrate that ARL4C recruits IQGAP1 and MMP14 to membrane protrusions are required. In addition, several controls would be needed throughout, to enhance to clinical and in vivo data sets. Specific requirements are as follows:

1) While ARL4C, IQGAP1 and MMP14 are located to protrusions, where they are clearly involved in matrix degradation, the loss of protein distribution to the protrusions (with ARL4C loss) could very well be due to the reduced abundance of protrusions associated with reduced migration and invasion of the cells. While a role for ARL4C in cell invasion is clearly demonstrated, the conclusion that ARL4C's "unique distribution to membrane protrusions is required for cancer cell invasion" is not supported by the data as presented.

This could be improved on by a variety of experiments. For example:

– All images of protrusions should include F-actin labeling to define protrusions. If the authors are arguing that ARL4C serves to recruit IQGAP1 and MMP14 to protrusions, then they need to show actin protrusions lack IQGAP1 and MMP14 when ARLC4 is knocked out.

– Studies that better demonstrate the role of protrusions in productive and/or directional migration are required. For example, the morphology of PDAC cells in the microfluidic assay suggests that the "cell protrusions" may in fact be focal adhesions/anchor points. Time-lapse imaging to show that these protrusions form via new offshoots perpendicular to the side of a cell versus a simple "zig-zag" motile pattern would be more convincing. Furthermore, time-lapse imaging of cells as they migrate towards the FBS media through the collagen matrix would enable one to determine if cell membrane protrusions are productive and have purposeful direction towards FBS. Hence, studies are needed that better characterize the direction and type of protrusions that are made.

– While ARLC4 knockouts are used for the localization studies, knock downs are used to study invasion. Both methods should be used for both assays.

– Please include a more thorough description of Mass Spec results so that the strength of the ARLC4-IQGAP1 interaction can be better appreciated.

– The 3D gel invasions demonstrate more cells in the control ASO migrating towards the FBS media, but shows there are also fewer cells in the ARLC4C ASO cells. Furthermore, there are more apoptotic bodies in the ARL4C ASO treatment cells. This suggests that the amount of siRNA/transfection agent used is toxic and interfering with migration and viability. An irrelevant binding control ASO should be used as well as a positive control ASO.

2) While ARLC4 shows promise as a target for PDAC, the preclinical studies should be improved, to strengthen correlations made and to ensure all controls are present in the in vivo assays. This could be improved on by a variety of experiments and/or edits. For example:

– Please include a table showing all patient information, as well as what the staining was for each patient.

– The images shown in Figure 1 suggest that ARLC4 protein levels in cancer cells is rather binary; however, the expression data suggests that this is not the case. This should be addressed, with a more thorough scoring scheme as well as analysis of the extent to which protein levels correlate with transcript levels. In particular, this is important when setting the cutoffs for survival curves, as an example.

– As presented in the manuscript, the clinical correlations made between IQGAP1 and ARL4C are not particularly well supported. The study would benefit from a more thorough correlational analysis as was shown for IQGAP1 and MMP14 and MMP14 and ARLC4 (supplemental 1).

– For the in vivo experiments, it is important that the extent of ARL4C knock down is established at the level of protein and transcript. In addition, the gene expression alterations should be shown in a table format. The effect of ASO1316 on ARL4C levels should also be shown in Figure 2G.

– In Figure 7H, the ASO control lesion reveals no tumor cells within the LYVE-1-delineated lumen. However, there is also a lack of tumor cells around this particular ROI. A more controlled analysis would evaluate all lymphatics that have the same amount of tumor cells surrounding them, and to look for lack of cells within the lymphatic lumen, if this is to be consistent with ASO's proposed inhibitory activity on cell invasion (and not cell proliferation or cytotoxicity).

*Reviewer #1:*

The current manuscript describes a functional relationship between ARL4C, IQGAP1 and MMP14 in pancreatic cancer cells; such ARLC4 is up-regulated in response to MAPK and Wnt signaling and colocalizes with IQGAP1 and MMP14 at membrane protrusions. This was associated with an increase in cellular invasion. Finally, the authors demonstrated that systemically delivered antisense oligonucleotides targeting ARL4C may prevent pancreatic cancer metastasis. Overall, this study reveals a hitherto undescribed mechanism for the regulation of pancreatic cancer cell invasion. The in vitro cell biology experiments are thorough and generally convincing. However, ties to clinical correlates as well as the in vivo metastasis assays could be improved upon.

Please include a table showing all patient information, as well as what the staining was for each patient.

The images shown in Figure 1 suggest that ARL4C protein levels in cancer cells is rather binary; however the expression data suggests that this is not the case. This should be addressed, with a more thorough scoring scheme as well as analysis of the extent to which protein levels correlate with transcript levels. In particular, this is important when setting the cutoffs for survival curves, as an example.

As presented in the manuscript, the clinical correlations made between IQGAP1 and ARL4C are not particularly well supported. The study would benefit from a more thorough correlational analysis as was shown for IQGAP1 and MMP14 and MMP14 and ARL4C (supplemental 1).

Please include a more thorough description of mass spec results so that the strength of the ARL4C-IQGAP1 interaction can be better appreciated.

For the in vivo experiments, it is important the extent of ARL4C knock down is established at the level of protein and transcript. In addition, the gene expression alterations should be shown in a table format.

*Reviewer #2:*

The authors undertook to assess the role of ARL4C-IQGAP1-MMP14 signaling downstream of Ras in pancreatic cancer progression and metastasis. Strengths are the comprehensive analysis of ARL4C in migration and 3D gel invasion, matrix degradation, and expression in tumors including mutational analysis that defines ARL4C domains that mediate these interactions and functional roles. Extension of these studies to use of anti-sense oligonucleotide to target ARL6 and particularly the in vivo data showing tumor regression are very interesting. A weakness of the study is the extensive analysis of protein localization to membrane protrusions, used to support the conclusion that ARL4C recruits IQGAP1 and MMP14 to protrusions. While these three proteins are located to protrusions, where they are clearly involved in matrix degradation, the loss of protein distribution to the protrusions could very well be due to the reduced abundance of protrusions associated with reduced migration and invasion of the cells. While a role for ARL4C in cell invasion is clearly demonstrated, the conclusion that ARL4C "unique distribution to membrane protrusions is required for cancer cell invasion" is not sufficiently supported by the data.

Figure 2G. The effect of ASO1316 on ARL4C expression is not shown.

Figure 2H. Not clear what the line graphs are showing.

Figure 3. All images of protrusions need to include F-actin labeling to define protrusions. If the authors are arguing that ARL4C serves to recruit IQGAP1 and MMP14 to protrusions, then they need to show actin protrusions lacking these proteins, not just count how many densities they observe. More likely, ARL4C and IQGAP1 are required for protrusion formation, something that is actually shown in Figure 3L for IQGAP1 knockdown, although not for ARL4C.

Figure 3I this is not clear.

Figure 3J. What is the relationship between low/high IQGAP1 and low /high ARL4C groups oin PDAC patient survival?

Figure 4. Show effect of ARL4C knockout on protrusions (E,F) but then resort to ASO to study invasion. What is the invasive capability of the KO cells? Same in Figure 5 G, H.

Figure 6. Not at all clear what the images are showing or what the merge is showing.*Reviewer #3:*

Kikuchi, Hamada, and colleagues continue their work on ARL4C and its role in promoting metastasis in pancreatic cancer by showing it acts by interacting with IQGAP1 which then interacts with MMP14. When this occurs, the entire complex is directed to membrane protrusions in pancreatic cancer cell lines. This is initiated by PIP3 and not PIP2 activation. Anti-sense oligos specific for ARL4C results in decreased lymphatic metastasis in a mouse model of pancreatic cancer, which is a surprising finding given that this is a transcriptionally based therapy. These pre-clinical results combined with their findings of elevated ARL4C in various PDAC tissue sections offer a novel opportunity to halt PDAC metastasis by antagonizing cell membrane protrusions caused by this ARL4C+IQGAP1+MMP14 complex.

Strengths:

1. If the main finding of this work is that cell protrusions are key for invasion/metastasis and ARL4C is a key protein involved in PDAC, then this is a very interesting and worthy paper. This is because the concept of a cell based organelle/feature that promotes metastasis is more likely to occur than a series of intricate molecular pathways. This concept is agnostic to any pathway involved and therefore the complex (ARL4C-IQGAP1-MMP14) observed here may or may not be broadly applicable to other cancers, but it is at least offers some molecular insights as to how it may happen.

2. PIP2/3 activation is a highly plausible activation pathway because it can work parallel to other classic outside/inside pathways such as beta1integrin activation that also can occur in cancer cells.

3. The IHC data strongly supports expression of ARL4C in perineural lesions, which is a strong risk factor for OS and HR in pancreatic cancer.

Suggestions for additional experiments:

1. Time-lapse imaging of cells as they migrate towards the FBS media through the collagen matrix would be helpful. It's not clear if the cell membrane protrusions are productive and have purposeful direction towards FBS.

2. The use of patient-derived PDX material would have added further momentum towards this idea that interrupting the ARL4C-IQGAP1-MMP14 complex leads to decreased cell membrane protrusion formation.

3. The use of organoids to understand the impact of these transgenes/mutations on cell membrane protrusions would have been closer to what is observed pre-clinically and clinically.

4. PDAC has a tremendous amount of fibrosis and is not a heavily vascularized tumor. Understanding the efficacy of ASO extravasation into tumors and how it precludes lymphatic metastasis would be of broad interest.

Open-ended questions:

1. The microfluidics chamber to assess uni-directional invasion by PDAC cells is innovative but the morphology of these cells suggests that the "cell protrusions" observed resemble focal adhesions/anchor points. Time-lapse imaging to show that these protrusions form via new offshoots perpendicular to the side of a cell versus a simple "zig-zag" motile pattern would have been convincing.

2. Only 20% of cells portray these IQGAP1-ARL4C rich protrusions. What is the metastatic efficiency like if these cells are removed from the total pool of cells within the tumor? Would there still be metastatic colony formation?

3. What are the cellular/biophysical barriers for PDAC cells as they intravasate into perineural space? Does this truly require breakdown of basement membrane or is another type of matrix/cellular barrier present? Such as myelin, fibroblasts, etc.?

4. What is the expression level of ARL4C in pre-PDAC lesions? Such as in PanIN2/3? What is/are the pioneer factors that induce ARL4C expression leading to PDAC? Hypoxia?

I have some concerns/comments regarding some of the findings and what else is missing.

1. Lian and Mulligan (Oncogene 2020) showed that "invasive processes" also contribute to perineural and neural invasion. These were driven by RET kinase activity and subsequent Src kinase activity. RET also needs to be analyzed in the IHC experiments since this is first published description of protrusions/invadopodia involved in perineural invasion in PDAC.

2. The 3D gel invasions demonstrate more cells in the control ASO migrating towards the FBS media but there are also fewer cells in the ARLC4C ASO cells. Furthermore, there are more apoptotic bodies in the ARL4C ASO treatment cells. This suggests that the amount of siRNA/transfection agent used is toxic and interfering with migration and viability. An irrelevant binding control ASO should be used as well as a positive control ASO.

3. "ARL4C and IQGAP1 were shown to accumulate to membrane protrusions at endogenous level in S2-CP8 and PANC-1 cells" the inset of the cell chosen doesn't appear to be a membrane protrusion, it may appear more as a focal adhesion anchorage point of the cell as it moves in that direction or away from that point. The same could be said for Figure 2C (Supplement#2 for Figure 3).

4. The cell protrusions formed by cells in the microfluidics chamber are of a radial projection. Do the authors contend that the cell protrusions form regardless of direction? What is the purpose or effectiveness of this kind of protrusion formation radial as opposed to the side of the cell facing the FBS?

5. In Figure 6I, there is a PanII lesion (large) that has abundant IQGAP expression and a minor amount of ARL4C protein expression. However, there is minimal MMP14 expression, save for some puncta. This suggests that ARC4C recruitment to IQGAP does not necessarily lead to MMP14 co-localization. Hence, is MMP14 a more important factor in the proposed mechanism than ARL4C and IQGAP?

6. What is the function of the cortical compartmentalization of ARL4C and IQGAP? (signal that is on the sides of the cells rather than the focal adhesions/cell protrusions)

7. Is there any impact on siRNA KD of IQGAP/ARL4C on protrusion formation as analyzed in Figure 3L when analyzed on the cells shown in Figure 4B (which only shows accumulation of ARL4C at protrusions and not if there is a change in the total number of protrusions).

8. Many of the experiments rely on overexpression of MMP14-GFP. Are the same results observed (Figure 6) when de novo MMP14 levels are evaluated?

9. Figure 7H is curious to me. The ASO control lesion reveals no tumor cells within the LYVE-1 lumen. However, there is also a lack of tumor cells around this particular ROI. A more controlled analysis would evaluate all lymphatics that have the same amount of tumor cells surrounding it (human mitochondria stain) and to look for lack of cells within the lymphatic lumen if this is to be consistent with the ASO's proposed inhibitory activity on cell invasion and not cell proliferation or cytotoxicity.

[Editors' note: further revisions were suggested prior to acceptance, as described below.]

Thank you for resubmitting your work entitled "Recruitment of KRAS downstream target ARL4C to membrane protrusions accelerates pancreatic cancer cell invasion" for further consideration by *eLife*. Your revised article has been evaluated by Erica Golemis (Senior Editor) and a Reviewing Editor.

The manuscript has been improved but there are some remaining issues that need to be addressed, as outlined below:

The study was revised and was improved; however, the conclusion that ARL4C recruits IQGAP1 and MMP14 to protrusions is still not adequately supported. Figures relating to these structures should define not only the presence of the protein of interest in protrusions but also the actin labeling and number of protrusions per cell. If these protrusions are defined by the absence of focal adhesions, then focal adhesion protein expression in these structures should also be shown. Alternatively, the focus in the manuscript on ARL4C regulation of these poorly defined membrane protrusions should be reduced and emphasis placed on regulation of cancer cell migration, invasion and invadopodia formation (on collagen) in addition to the in vivo data.

*Reviewer #1:*

Previous reviews were responded to well, with the addition of new data/and or more thorough descriptions of methodology and data.

I am satisfied with this revision.

*Reviewer #2:*

The manuscript is interesting and defines a role for ARL4C in pancreatic cell migration, invasion and tumor progression. I still however have serious issues about the data on membrane protrusions that is presented, and indeed the definition of membrane protrusions that I outline here:

In the abstract: ARL4C recruited IQGAP1 and its downstream effector, MMP14, to membrane protrusions. Specific localization of ARL4C, IQGAP1, and MMP14 was the active site of invasion, which induced degradation of the extracellular matrix.

From the text: Membrane protrusions were defined as actin-based structure of which length is longer than 10 μm and diameter is shorter than 10 μm. Also, cell protrusions were not stained with focal adhesion marker (Figure 2—figure supplement 1F).

What are these protrusions? Leading edge or trailing edge. Absence of focal adhesions and images from 2F and others showing focal adhesion and actin-rich lamellipodia at the opposing end of the cell suggests that they are trailing edge. If so how to reconcile with the regulation of invadopodia which are necessarily leading edge? While well-defined focal adhesions are not present in invadopodia, invadopodia still retain integrin-based matrix adhesions. Pseudopodia and lamellipodia in cancer cells plated on cover slips all contain focal adhesions. If the focal adhesion free membrane protrusions that the authors are studying are indeed protrusive structures analagous to the invadopodia they study in cells plated on collagen, then they must show this using live cell imaging. If they are protrusive structures, how to reconcile the absence of focal adhesions proteins with the extensive literature defining a role of focal adhesions/contacts/ matrix adhesions in pseudopodia protrusion and tumor cell migration and invasion? If these protrusions are retracting trailing edge structures, not analogous to invadopodia, what is their role in invasion and migration? Overall, these protrusions and their role in cell migration and invasion need to be better defined.

Specifically, protrusions shown in untreated WT have clear actin densities – but protrusions in treated cells lacking enrichment of a protein of interest (ARL6, IQGAP1, MMP-14 …) do not. This raises the question as to whether the targeted treatments that inhibit migration and invasion are also preventing formation of protrusions? Also, that quantification is based on number of "cells presenting protein enriched protrusions" is troubling. Do the various treatments alter the number of protrusions per cell? Do they alter the actin density of the protrusions as seems evident from some of the data presented. If so are the treatments altering the nature of the protrusion or ARL4C recruitment to the protrusions?

For example, PI3K is well known to be required for actin-dependent pseudopod protrusion – so the presentation that LY294002 prevents ARL4C and IQGAP1 accumulation at "protrusions" is not surprising. However, it raises serious questions as to what exactly are the protrusions that are being measured.

Without defining exactly what these membrane protrusions and a clear demonstration that the focal adhesion-free protrusions on glass are analagous to protrusive invadopodia, the idea that "Recruitment of KRAS downstream target ARL4C to membrane protrusions accelerates pancreatic cancer cell invasion" is interesting but not supported by the data presented and does not provide a clear mechanistic understanding of the role of ARL4C in cancer invasion. At best there is a correlation of association of ARL4C and as yet to be defined membrane protrusive structures.

Here is a list of protrusion data in the paper. Each of these figures should define not only the presence of the protein of interest in protrusions but also the actin labeling and number of protrusions per cell. If these protrusions are defined by the absence of focal adhesions then focal adhesion protein expression in these structures should also be shown. Alternatively, the focus in the manuscript on ARL4C regulation of these poorly defined membrane protrusions should be reduced and emphasis placed on regulation of cancer cell migration, invasion and invadopodia formation (on collagen) in addition to the in vivo data.

Figure 2K ARL4C ASO ◊ reduced number of collagen positive protrusions. What about actin protrusions?

Figure 2 Supp 1 The T27N ARL4C-GFP mutant is said not to accumulate at protrusions – but does it prevent protrusion formation? need to also quantify #actin protrusions

Supp1H,J – ARL4C ASO and KO reduce protrusions by about 5-10% but invasion by 80-90%

Figure 2 Supp 2C,E – ARL4C-GFP does not induce invadopodia in BxPC3 cells but does induce protrusions.

Figure 3M IQGAP siRNA reduces actin protrusions, even the increased number induced by OE of ARL4C-GFP

Figure 4B just look at ARL4C protrusions – are there actin-rich protrusions?

Figure 4E WT protrusions have clear actin densities – but others do not – based on what are these structures considered to be pseudopodia/ lamellipodia and not trailing edges of cells. There is a clear lack of actin densities in mock G2A DPBR cells and the protrusions selected in the WT and Mock cells are elongated protrusions at the opposite end of the cells from actin-rich lamellipodia and pseudopodia.

Figure 4 Supp 1B – ARL4C ASO reduces IQGP-1 accumulation to protrusions – what about #protrusions; what about actin accumulation in protrusions? In A looks as though ARL4SO is reducing actin accumulation in protrusions. Same in Figure 6 Supp 1C – mmp-14 down so is actin.

Figure 5 B – PI3K is well known to be required for actin-dependent pseudopod protrusion – so the presentation that LY294002 prevents Arl4C and IQGAP1 accumulation at "protrusions" is not surprising. However, it raises serious questions as to what exactly are the protrusions that are being measured. This has to be clarified – are they actin-rich? Are they leading or trailing edge?

Figure 5F, G again are there actin -rich protrusions under the treated conditions? Are the authors measuring recruitment of ARL6 and IQGAP1 to protrusions – or are they measuring that treatments are disrupting protrusions (i.e. pseudopodia and invadopodia)? The latter is more likely and consistent with their migration and invasion data.

Figure 6 B, C,D same as above for MMP-14.

Figure 6 Supp1G, H – WT vs ARL4C KO and WT vs ∆C show similar minimal trends and one is significant and one is not – is this supposed to be a real effect?

---

## [Author Response]

Essential revisions:A number of issues were raised that need to be addressed with experimentation, prior to publication. Specifically, studies that more concretely demonstrate that ARL4C recruits IQGAP1 and MMP14 to membrane protrusions are required. In addition, several controls would be needed throughout, to enhance to clinical and in vivo data sets. Specific requirements are as follows:1) While ARL4C, IQGAP1 and MMP14 are located to protrusions, where they are clearly involved in matrix degradation, the loss of protein distribution to the protrusions (with ARL4C loss) could very well be due to the reduced abundance of protrusions associated with reduced migration and invasion of the cells. While a role for ARL4C in cell invasion is clearly demonstrated, the conclusion that ARL4C's "unique distribution to membrane protrusions is required for cancer cell invasion" is not supported by the data as presented.This could be improved on by a variety of experiments. For example:– All images of protrusions should include F-actin labeling to define protrusions. If the authors are arguing that ARL4C serves to recruit IQGAP1 and MMP14 to protrusions, then they need to show actin protrusions lack IQGAP1 and MMP14 when ARLC4 is knocked out.

We agree to the comment that membrane protrusions should be shown by F-actin labeling. ARL4C and IQGAP1 are involved in the regulation of actin cytoskeleton. Two dimensional (2D) experiments are suitable for intracellular localization of proteins. In this study we defined membrane protrusions as actin-based structure of which length is longer than 10 μm and diameter is shorter than 10 μm. ARL4C KO affected the structure of protrusions slightly and decreased numbers of cells with protrusions by about 10%. However, IQGAP1 and MMP14 disappeared in most of protrusions where ARL4C is knocked out. Moreover, in Figure 4E not only ARL4C KO cells but also all the KO cells expressing various ARL4C mutants show actin protrusions with or without IQGAP1 accumulation. The results are described in the text (page 6, lines 15 and 16, page 11, lines 12 and 13, page 13, lines 7 through 10) and shown in Figure 4E, Figure 4—figure supplement 1A and B, and Figure 6—figure supplement 1C and D.

– Studies that better demonstrate the role of protrusions in productive and/or directional migration are required. For example, the morphology of PDAC cells in the microfluidic assay suggests that the "cell protrusions" may in fact be focal adhesions/anchor points. Time-lapse imaging to show that these protrusions form via new offshoots perpendicular to the side of a cell versus a simple "zig-zag" motile pattern would be more convincing. Furthermore, time-lapse imaging of cells as they migrate towards the FBS media through the collagen matrix would enable one to determine if cell membrane protrusions are productive and have purposeful direction towards FBS. Hence, studies are needed that better characterize the direction and type of protrusions that are made.

Focal adhesions are defined as paxillin-containing, multi-protein structures that form mechanical links between intracellular actin bundles and the extracellular substrate. “Cell protrusions” that we pointed out throughout this study were not stained with anti-paxillin antibody. Paxillin was clearly observed in cell adhesion sites, which were different from cell protrusions. Therefore, they are unlikely to be focal adhesions. The results are described in the text (page 6, lines 16 and 17) and shown in Figure 2—figure supplement 1F.

Using time-lapse imaging we revealed that the protrusions are extended in the direction of cell invasion. We also calculated the angle of protrusions to the direction of cell movement towards FBS. Most of them were located in the angle of -45 to +45 degrees in the polar histogram. These results suggest that membrane protrusions play a role in purposeful direction. The results are described in the text (page 7, lines 15 through 18) and shown in Figure 2I and Figure 2-video 2.

– While ARLC4 knockouts are used for the localization studies, knock downs are used to study invasion. Both methods should be used for both assays.

According to the comments, we performed the localization assays with ARL4C knockdown and the invasion assays with ARL4C knockout. Depletion of ARL4C by either method showed the same phenotypes. The results are described in the text (page 7, lines 5 through 7, page 11, lines 13 and 14, page 13, lines 7 through 10) and shown in Figure 2—figure supplement 1J, Figure 4—figure supplement 1B, and Figure 6—figure supplement 1D.

– Please include a more thorough description of Mass Spec results so that the strength of the ARLC4-IQGAP1 interaction can be better appreciated.

We apologize for our poor description. In Supplementary File 1 Table 2-source data 1, the numbers of peptides recognized by mass spectrometry for each protein were described with available information.

– The 3D gel invasions demonstrate more cells in the control ASO migrating towards the FBS media, but shows there are also fewer cells in the ARLC4C ASO cells. Furthermore, there are more apoptotic bodies in the ARL4C ASO treatment cells. This suggests that the amount of siRNA/transfection agent used is toxic and interfering with migration and viability. An irrelevant binding control ASO should be used as well as a positive control ASO.

As a control ASO, we used randomized nucleotides for the sequence of ARL4C ASO. As already shown in Figure 2—figure supplement 1A. ARL4C ASO did not affect pancreatic cancer cell growth compared with control ASO. In addition, both ASOs did not induce cell death, which is assessed by propidium iodide (PI) staining. Thus, ASOs used in this study are not toxic for pancreatic cancer cells. The results are described in the text (page 6, lines 5 and 6) and shown in Figure 2—figure supplement 1C.

The reviewer criticized that there are few cells in ARL4C ASO-treated cells in the 3D gel invasion assay. At 0 time we placed the same numbers of cells treated with control and ARL4C ASOs in the starting position, and counted the numbers of invading cells after 72 h. Under the conditions, ARL4C ASO inhibited invasion ability of pancreatic cancer cells. The results are described in the text (page 7 lines 3 through 6) and shown in Figure 2—figure supplement 1I.

2) While ARLC4 shows promise as a target for PDAC, the preclinical studies should be improved, to strengthen correlations made and to ensure all controls are present in the in vivo assays. This could be improved on by a variety of experiments and/or edits. For example:– Please include a table showing all patient information, as well as what the staining was for each patient.

According to the comment, all patients (57 cases) information was described in Supplementary File 1 Table 1-source data 1. This statement is described in the text (page 4, lines 18 through 20).

– The images shown in Figure 1 suggest that ARLC4 protein levels in cancer cells is rather binary; however, the expression data suggests that this is not the case. This should be addressed, with a more thorough scoring scheme as well as analysis of the extent to which protein levels correlate with transcript levels. In particular, this is important when setting the cutoffs for survival curves, as an example.

In Figure 1A and B, ARL4C expression was considered high when the total area of the tumor stained with anti-ARL4C antibody exceeded 5%. Under the definition, 82% of cases were judged as high expression. According to the comments, the ARL4C expression levels were scored as continuous variables based on the percentages of ARL4C-staining areas to total tumor areas. The detailed scores were described in Supplementary File 1 Table 1-source data 1.

It is hard to assess which protein levels correlate with transcription levels in public datasets. Therefore, dataset cases were separated into *ARL4C* high and low expression groups based on the top 75% of mRNA values of *ARL4C*, and 131 of 174 pancreatic cancer cases were classified as a high expression group, of which proportion was similar to that of immunohistochemical study for ARL4C. These statements are described in the text (page 4, lines 20 through 23, page 5, lines 7 through 9) and shown in Figure 1D.

– As presented in the manuscript, the clinical correlations made between IQGAP1 and ARL4C are not particularly well supported. The study would benefit from a more thorough correlational analysis as was shown for IQGAP1 and MMP14 and MMP14 and ARLC4 (supplemental 1).

As pointed out, the overall survival of PDAC patients who were double positive for ARL4C and IQGAP1 tended to be worse but not statistically significant (Figure 3—figure supplement 2J). According to the comment, the correlation of the expression between ARL4C and IQGAP1 was examined in the TCGA dataset. The dataset showed that expression of *ARL4C* mRNA in pancreatic cancer patients is positively correlated with that of *IQGAP1* mRNA. The results are described in the text (page 10, lines 3 through 5) and shown in Figure 3K.

– For the in vivo experiments, it is important that the extent of ARL4C knock down is established at the level of protein and transcript. In addition, the gene expression alterations should be shown in a table format. The effect of ASO1316 on ARL4C levels should also be shown in Figure 2G.

The effects of ARL4C ASO on ARL4C levels in pancreatic cancer cells in vitro were already demonstrated in Figure 2—figure supplement 1B of the original manuscript. We have also reported that ARL4C ASO-1316 suppresses the levels of *ARL4C* mRNA and protein in hepatic cancer HLE cells and inhibits in vivo tumor formation induced by HLE cells (Harada, T. et al. Mol. Cancer Ther. 2018). Thus, the efficacy of ARL4C ASO-1316 was shown in different types of cancer cells.

The mRNA and protein levels of ARL4C in the pancreatic tumors obtained from the mice treated with control or ARL4C ASO were analyzed by immunohistochemistry, qPCR, and Western blotting. ARL4C ASO-1316 indeed reduced ARL4C expression in tumor lesions. The results are described in the text (page 14, lines 17 and 18, page 15, lines 1 and 2) and shown in Figure 7C, 7H, and Figure 7—figure supplement 1B.

The gene expression alterations in RNA-seq analysis based on log2 (fold change) > 1.0 or log2 (fold change) < -1.0 and FDR < 0.1, which was affected by ARL4C ASO treatment, was shown in Figure 7-data source 2.

– In Figure 7H, the ASO control lesion reveals no tumor cells within the LYVE-1-delineated lumen. However, there is also a lack of tumor cells around this particular ROI. A more controlled analysis would evaluate all lymphatics that have the same amount of tumor cells surrounding them, and to look for lack of cells within the lymphatic lumen, if this is to be consistent with ASO's proposed inhibitory activity on cell invasion (and not cell proliferation or cytotoxicity).

First of all, in the orthotopic transplantation model, lymphatic vessels of the host were located around the tumor mass implanted. According to the comment, we carefully checked many lymphatics and confirmed again that tumor cell numbers surrounding lymphatic vessels are decreased by ARL4C ASO. We believe this makes sense, because ARL4C ASO reduces the numbers of tumor cell that leave the primary site due to inhibition of cell invasive ability. Since ARL4C ASO did not affect primary tumor sizes, it is consistent with our hypothesis that ARL4C is involved in invasion but not in proliferation of pancreatic cancer cells. These statements are described in the text (page 15, lines 4 through 6). We also confirmed that ARL4C ASO does not induce cell death as shown in Figure 2—figure supplement 1C.

Reviewer #2:Figure 2H. Not clear what the line graphs are showing.

We apologize for our insufficient description. Fluorescence intensities of ARL4C-tdTomato in the edges of cell protrusions and cytoplasm, which are indicated by white arrowheads and yellow closed circles (20 μm away from the tip of protrusion), respectively, were measured over time, and then the intensities were plotted as a function of time. The results indicate that ARL4C is dynamically appeared and disappeared in the protrusions, but it did not accumulate in the cytoplasm. The statements are described in the text (page 7, lines 11 through 15).

Figure 3I this is not clear.

We examined the IQGAP1 expression levels in pancreatic cancer immunohistochemically and found that 31 cases of 57 PDAC patients (54%) highly expressed IQGAP1 as shown in Figure 3I and Supplementary file 1 Table 3. The left image of original Figure 3I indicates that both IQGAP1 and ARL4C were expressed in the serial section. The right graph showed that IQGAP1 high expression cases are observed in 27 cases of 47 ARL4C high expression cases, and in 4 cases of 10 ARL4C low expression cases. However, these data do not show significant results, probably due to limited numbers of cases. Therefore, we removed the image showing ARL4C expression and showed IQGAP1 expression in tumor lesions and non-tumor regions in Figure 3I. Instead, we analyzed the relationship between ARL4C and IQGAP1 using public datasets. The statements are described in the text (page 10, lines 3 through 5) and the results are shown in Figure 3K.

Figure 3J. What is the relationship between low/high IQGAP1 and low /high ARL4C groups oin PDAC patient survival?

The overall survival of PDAC cases who highly expressed both ARL4C and IQGAP1 was not statistically significant compared with that of the cases with high ARL4C and low IQGAP1, probably due to the limited case numbers (Figure 3—figure supplement 2J). Although we examined only 57 PDAC cases, these are rare cases without the use of chemotherapy and benefit for immunohistochemically staining, because chemotherapy is usually administered in pancreatic cancer patients prior to operation. I would appreciate it if the reviewer could understand the reasons of the limited case numbers that we used.

Therefore, we analyzed the relationship between ARL4C and IQGAP1 expression on patient survival using public datasets. The results demonstrated that overall survival was significantly decreased in the order of low ARL4C/low IQGAP1, high ARL4C/low IQGAP1, and high ARL4C/high IQGAP1, although the result of low ARL4C/high IQGAP1 could not conclude because of the small sample size. The results are described in the text (page 10, lines 10 through 14) and shown in Figure 3—figure supplement 2K.

Figure 4. Show effect of ARL4C knockout on protrusions (E,F) but then resort to ASO to study invasion. What is the invasive capability of the KO cells? Same in Figure 5 G, H.

To respond to the reviewer’s comment, we showed that knockdown and knockout of ARL4C little affect the extension of membrane protrusions. The results are described in the text (page 6, lines 22 through 24) and shown in Figure 2—figure supplement 1H. Then we demonstrated that depletion of ARL4C by either method inhibits invasion ability. The results are described in the text (page 7, lines 5 through 7) and shown in Figure 2F and Figure 2—figure supplement 1J. In addition, we examined the effect of ARL4C knockout and knockdown on localization of IQGAP1 and MMP14 and found that depletion of ARL4C suppressed their localization to the tip of membrane protrusion. The results are described in the text (page 11, lines 12 through 14; page 13, lines 7 through 10) and shown in Figure 4—figure supplement 1A and B and Figure 6—figure supplement 1C and D.

Thus, we would like to emphasize that invasion ability and localization of IQGAP1 and MMP14 at the membrane protrusions are inhibited by knockdown and KO by ARL4C, but the structure of the protrusions is little changed.

Figure 6. Not at all clear what the images are showing or what the merge is showing.

We believe that the reviewer pointed out Figure 6I. In this experiment, we tried to show that ARL4C, IQGAP1, and MMP14 are simultaneously expressed in invading pancreatic cancer cells by staining the serial section of human specimen using triple immunofluorescence imaging assay. Let me firstly explain our interpretation of Figure 6I and Figure 6—figure supplement 1H of the original manuscript

In Figure 6I of the original manuscript, PanIN lesion was shown at the upper half of the figure. The cell in the yellow dashed boxes was an invasive PDAC cell. ARL4C and MMP14 were expressed more highly in invasive cancer cells rather than in PanIN lesions, although IQGAP1 was thoroughly expressed in tumor lesions including PanIN areas. Merged images on the right bottom indicated that invasive cancer cells express the three proteins simultaneously. Low-power images were shown in Figure 6—figure supplement 1H of the original manuscript, which demonstrates a group of cells invaded the surrounding interstitial tissues, and concurrently expressed three proteins. However, these statements were not well described.

Therefore, we replaced the figures in the revised manuscript and again emphasized that ARL4C, IQGAP1, and MMP14 are expressed together in invading cells. The results are described in the text (page 14, lines 3 through 8) and shown in Figure 6I and Figure 6—figure supplement 1J.

Reviewer #3:Suggestions for additional experiments:1. Time-lapse imaging of cells as they migrate towards the FBS media through the collagen matrix would be helpful. It's not clear if the cell membrane protrusions are productive and have purposeful direction towards FBS.2. The use of patient-derived PDX material would have added further momentum towards this idea that interrupting the ARL4C-IQGAP1-MMP14 complex leads to decreased cell membrane protrusion formation.3. The use of organoids to understand the impact of these transgenes/mutations on cell membrane protrusions would have been closer to what is observed pre-clinically and clinically.

We agree to the reviewer’s comment that the experiments with PDX materials and organoids would strengthen our model in vivo. However, since to use these materials is beyond the aim of this study, we would like to perform these experiments in the future.

4. PDAC has a tremendous amount of fibrosis and is not a heavily vascularized tumor. Understanding the efficacy of ASO extravasation into tumors and how it precludes lymphatic metastasis would be of broad interest.

Yes, as the reviewer said, fibrosis of pancreatic cancer blocks the transport of medicine from the vessels to cancer cells. Honestly, we have not yet completely understood how ASO reaches to the tumors after subcutaneous administration. Since the orthotopic implantation tumors do not have a large amount of fibrosis around tumors, it would be good to use the KPC (Kras^G12D^, p53^mutant^, and Cre) pancreatic cancer mouse model for evaluation of ARL4C ASO as future experiments.

Open-ended questions:1. The microfluidics chamber to assess uni-directional invasion by PDAC cells is innovative but the morphology of these cells suggests that the "cell protrusions" observed resemble focal adhesions/anchor points. Time-lapse imaging to show that these protrusions form via new offshoots perpendicular to the side of a cell versus a simple "zig-zag" motile pattern would have been convincing.2. Only 20% of cells portray these IQGAP1-ARL4C rich protrusions. What is the metastatic efficiency like if these cells are removed from the total pool of cells within the tumor? Would there still be metastatic colony formation?

As shown by time-lapse imaging in Figure 2-video 1 and Figure 2H, cell protrusions are dynamically formed. Therefore, there are some cells with protrusions and other cells without protrusions at one point. We demonstrated still images of cultured cells in most figures to show detailed localization of proteins clearly. We believe that most of cancer cells expressing ARL4C extend protrusions, to which ARL4C is localized, in tumors in vivo, and these cells could invade into the stroma and metastasize.

3. What are the cellular/biophysical barriers for PDAC cells as they intravasate into perineural space? Does this truly require breakdown of basement membrane or is another type of matrix/cellular barrier present? Such as myelin, fibroblasts, etc.?

Perineural invasion has a prevalence more than 70% of pancreatic cancer patients, and is associated with poor prognosis. However, the mechanisms underlying perineural invasion are poorly understood. Peripheral nerve structure consists of the endoneurium, perineurium, and epineurium. The endoneurium is the innermost layer and consists of nerve fibers, composed of axons surrounded by Schwann cells; it also contains blood vessels, resident macrophages and fibroblasts. The perineurium, which surrounds the endoneurium, is a layer of cylindrical cells tightly interconnected, forming a protective barrier. The epineurium is the outermost layer surrounding several nerve bundles; it includes an elastin and collagen sheath, blood and lymphatic vessels, resident macrophages, mast cells, and fibroblasts. Therefore, there would be several possible mechanisms that PDAC cells intravasate into perineural space. As the reviewer suggested, pancreatic cancer cells need to break epineurium and perineurium cells and degrades elastin, collagen, and myelin to damage endoperium. More studies are absolutely required for the understanding of whole picture of perineural invasion mechanisms. Since we do not have any data, I would like to refrain from discussing the mechanisms of perineural invasion.

4. What is the expression level of ARL4C in pre-PDAC lesions? Such as in PanIN2/3? What is/are the pioneer factors that induce ARL4C expression leading to PDAC? Hypoxia?

This is a great suggestion. According to the comment, we examined ARL4C expression in 26 cases of PanIN and found that 20 cases are positive for ARL4C, suggesting that ARL4C is expressed in precancerous stage of pancreatic cancer. The results are described in the text (page 5, lines 13 through 17) and shown in Figure 1—figure supplement 1C. These results are consistent with our recent observations that ARL4C is frequently expressed in atypical adenomatous hyperplasia, which is the possible precursor lesions and develops to lung adenocarcinoma Kimura, K. et al., Cancer Sci, 2020. Therefore, we need to examine the possibility of ARL4C expression of the early stage of cancer development more systematically.

So far, we have found that ARL4C is expressed downstream of the Wnt/β-catenin and Ras pathways in colon cancer, lung adenocarcinoma, and pancreatic cancer (Oncogene 2017, Oncotarget 2018, and this study). These pathways are activated by mutations of APC, β-catenin, and Ras. However, we also showed that ARL4C is expressed by de-methylation of 3’-UTR of the ARL4C gene in lung and oral squamous cell carcinoma (Oncotarget 2018). It would be interesting to search for another factor that induces ARL4C expression.

I have some concerns/comments regarding some of the findings and what else is missing.1. Lian and Mulligan (Oncogene 2020) showed that "invasive processes" also contribute to perineural and neural invasion. These were driven by RET kinase activity and subsequent Src kinase activity. RET also needs to be analyzed in the IHC experiments since this is first published description of protrusions/invadopodia involved in perineural invasion in PDAC.

We are currently investigating the functional relationship between the ARL4C pathway and other signaling pathways. We would like to examine RET expression in PDAC in the next project.

2. The 3D gel invasions demonstrate more cells in the control ASO migrating towards the FBS media but there are also fewer cells in the ARLC4C ASO cells. Furthermore, there are more apoptotic bodies in the ARL4C ASO treatment cells. This suggests that the amount of siRNA/transfection agent used is toxic and interfering with migration and viability. An irrelevant binding control ASO should be used as well as a positive control ASO.3. "ARL4C and IQGAP1 were shown to accumulate to membrane protrusions at endogenous level in S2-CP8 and PANC-1 cells" the inset of the cell chosen doesn't appear to be a membrane protrusion, it may appear more as a focal adhesion anchorage point of the cell as it moves in that direction or away from that point. The same could be said for Figure 2C (Supplement#2 for Figure 3).4. The cell protrusions formed by cells in the microfluidics chamber are of a radial projection. Do the authors contend that the cell protrusions form regardless of direction? What is the purpose or effectiveness of this kind of protrusion formation radial as opposed to the side of the cell facing the FBS?5. In Figure 6I, there is a PanII lesion (large) that has abundant IQGAP expression and a minor amount of ARL4C protein expression. However, there is minimal MMP14 expression, save for some puncta. This suggests that ARC4C recruitment to IQGAP does not necessarily lead to MMP14 co-localization. Hence, is MMP14 a more important factor in the proposed mechanism than ARL4C and IQGAP?

As pointed out by the reviewer, tumor lesions in Figure 6I of the original manuscript could be PanIN lesions. In this experiment, we tried to show that ARL4C, IQGAP1, and MMP14 are simultaneously expressed in invading pancreatic cancer cells by staining the serial section of human specimen using triple immunofluorescence imaging assay. Let me explain our interpretation. As suggested, PanIN lesion was shown at the upper half of the figure as indicated. The cell in the yellow dashed boxes was an invasive PDAC cell. ARL4C and MMP14 were expressed more highly in invasive cancer cells rather than in PanIN lesions, although IQGAP1 was thoroughly expressed in tumor lesions including PanIN areas. Merged images on the right bottom indicated that invasive cancer cells express the three proteins simultaneously. Low-power images were shown in Figure 6—figure supplement 1H of the original manuscript, which demonstrates a group of cells invaded the surrounding interstitial tissues, and concurrently expressed three proteins. However, these statements were not well described.

Therefore, we replaced the figures in the revised manuscript and again emphasized that ARL4C, IQGAP1, and MMP14 are expressed together in invading cells. The results are described in the text (page 14, lines 3 through 8) and shown in Figure 6I and Figure 6—figure supplement 1J.

6. What is the function of the cortical compartmentalization of ARL4C and IQGAP? (signal that is on the sides of the cells rather than the focal adhesions/cell protrusions)

ARL4C is localized to the plasma membrane through myristoylation. In addition, we found that the binding of PIP3 and polybasic region of ARL4C is required for the localization to cell protrusions, and PIP3 is enriched in protrusions rather than PIP2. Through the mechanism, IQGAP1 is recruited to cell protrusions. These are important findings in this project.

7. Is there any impact on siRNA KD of IQGAP/ARL4C on protrusion formation as analyzed in Figure 3L when analyzed on the cells shown in Figure 4B (which only shows accumulation of ARL4C at protrusions and not if there is a change in the total number of protrusions).

In this study we defined membrane protrusions as actin-based structure of which length is longer than 10 μm and diameter is shorter than 10 μm. Knockdown and KO of ARL4C affected the structure of protrusions slightly and decreased numbers of cells with protrusions by only about 10%, suggesting that ARL4C is not essential for extension of protrusions. The results are described in the text (page 6, lines 22 through 24) and shown in Figure 2—figure supplement 1H.

8. Many of the experiments rely on overexpression of MMP14-GFP. Are the same results observed (Figure 6) when de novo MMP14 levels are evaluated?

According to the comment, de novo MMP14 expression levels were examined. Endogenous MMP14 were predominantly present in the cytoplasm but hardly detected in the plasma membrane, because MMP14 is recycled between the plasma membrane and cytoplasm. The results are shown in Author response image 1. Therefore, we used MMP14-GFP to clearly show the localization of MMP14 to the plasma membrane.

**Author response image 1. sa2fig1:** Validation of anti-MMP14 antibody.S2-CP8 cells were stained with anti-MMP14 antibody and phalloidin.

9. Figure 7H is curious to me. The ASO control lesion reveals no tumor cells within the LYVE-1 lumen. However, there is also a lack of tumor cells around this particular ROI. A more controlled analysis would evaluate all lymphatics that have the same amount of tumor cells surrounding it (human mitochondria stain) and to look for lack of cells within the lymphatic lumen if this is to be consistent with the ASO's proposed inhibitory activity on cell invasion and not cell proliferation or cytotoxicity.

[Editors' note: further revisions were suggested prior to acceptance, as described below.]

Reviewer #2:What are these protrusions? Leading edge or trailing edge. Absence of focal adhesions and images from 2F and others showing focal adhesion and actin-rich lamellipodia at the opposing end of the cell suggests that they are trailing edge. If so how to reconcile with the regulation of invadopodia which are necessarily leading edge? While well-defined focal adhesions are not present in invadopodia, invadopodia still retain integrin-based matrix adhesions. Pseudopodia and lamellipodia in cancer cells plated on cover slips all contain focal adhesions. If the focal adhesion free membrane protrusions that the authors are studying are indeed protrusive structures analagous to the invadopodia they study in cells plated on collagen, then they must show this using live cell imaging. If they are protrusive structures, how to reconcile the absence of focal adhesions proteins with the extensive literature defining a role of focal adhesions/contacts/ matrix adhesions in pseudopodia protrusion and tumor cell migration and invasion? If these protrusions are retracting trailing edge structures, not analogous to invadopodia, what is their role in invasion and migration? Overall, these protrusions and their role in cell migration and invasion need to be better defined.

We appreciate Reviewer #2’s critique of our definition of “membrane protrusion”. We used the term “membrane protrusion” as the structure extending from cells. In the first revised manuscript (Figure 2—figure supplement 1G) we stated that paxillin, a core component of focal adhesion complex, is not detected in cell protrusions. According to the Reviewer’s critique, we carefully stained cells with anti-paxillin antibody repeatedly under the different blocking conditions and found that paxillin is detected in the structures (Figure 2—figure supplement 1F). In addition, other focal adhesion proteins such as phosphorylated paxillin, FAK, and phosphorylated FAK, and F-actin were also detected in the tips of the protrusions (Figure 2—figure supplement 1F). Therefore, the membrane protrusions that we defined are actin-based structures that contain the adhesion sites. These are described in the text (page 6, lines 18 through 22). We really thank to Reviewer #2’s appropriate comment.

Since the protrusions had focal adhesion sites and contacted the surrounding extracellular matrix, next we investigated the invasive properties of the protrusions. It is known that invadopodia are key structures for cancer cell invasion and have been extensively studied using some cells, including MDA-MB-231 (breast cancer cell) and SCC61 (head and neck squamous carcinoma cell). Therefore, we firstly explored invadopodia formation in pancreatic cancer cell lines. In my understanding, invadopodia are the unique structure which extend vertically from the cancer cell bottom to the extracellular matrix and the structure is detected on a gelatin-coated glass coverslip, using a well-known “invadopodia assay.” As shown in Figure 2—figure supplement 2A, invadopodia were detected in BxPC-3 but not S2-CP8 or PANC-1 cells. Therefore, we thought that pancreatic cancer cells (S2-CP8 and PANC-1 cells) that we used in this study do not form typical invadopodia from the ventral side of the cell body but can invade into extracellular matrix through other structures. This is the reason why we used the term “membrane protrusion” but not the term “invadopodia.” However, we found that invadopodia markers such as cortactin and ARPC2 localize to the tips of protrusions with ARL4C, suggesting that the protrusions might contribute to invasive phenotypes of pancreatic cancer cells and ARL4C functions there (Figure 2—figure supplement 2B).

It has been reported that cells lacking MMP14 display no defects in 2D proliferation or migration across collagen-coated surface, but their capacity to invade is severely impaired [J. Cell Biol. 167, 769, 2004]. The results resemble our findings obtained by ARL4C knockdown. In 3D gel invasion assay conditions, cancer cells clear their path using MMP14 degradative activity and the leading protrusions of invasive cancer cells have the capacity to degrade collagen fibers. Thus, the membrane protrusions that we defined are structurally and functionally similar to invadopodia in that ARL4C recruits MMP14 through IQGAP1 to the tip of the structures where the same component with invadopodia are localized and that cells with the protrusions can penetrate into the collagen gel in 3D collagen assay. Therefore, we referred to the protrusive structures as “invasive pseudopods.” Our results definitively show that ARL4C recruits MMP14 to the tips of invasive pseudopods to degrade the ECM. These are described in the text (page 7, lines 3 through 15).

In addition, cortactin and ARPC2 are also well-known marker for the leading edge of the cell (Nat. Rev. Mol. Cell Biol. 7, 713, 2006, Oncotarget, 7, 46142-46157, 2016). They localize to the tips of pseudopods and contribute to invasion of cells (Sci. Signal. 4, issue 159, pe6, 2011, J Cell Biol. 199, 527, 2012). “Invasive pseudopods” which we have defined show a clear localization of leading edge markers, indicating that the pseudopods are formed at the front of cells to keep cells move forward. This observation can also be confirmed in Figure 2I and Figure 2-video 2 and is the answer for Reviewer #2.

Specifically, protrusions shown in untreated WT have clear actin densities – but protrusions in treated cells lacking enrichment of a protein of interest (ARL6, IQGAP1, MMP-14 …) do not. This raises the question as to whether the targeted treatments that inhibit migration and invasion are also preventing formation of protrusions? Also, that quantification is based on number of "cells presenting protein enriched protrusions" is troubling. Do the various treatments alter the number of protrusions per cell? Do they alter the actin density of the protrusions as seems evident from some of the data presented. If so are the treatments altering the nature of the protrusion or ARL4C recruitment to the protrusions?

Another important critique of Reviewer #2 is whether ARL4C is involved in the formation of invasive pseudopods. As shown in experiments with depletion of ARL4C, the numbers of cells with invasive pseudopods were decreased slightly, while knockdown of ARPC2 which regulates formation of pseudopods as one of the components of Arp2/3 complex clearly reduced the number of pseudopods (Figure 2—figure supplement 2C-G). We tested whether ARL4C is involved in the presence of invadopodia markers or leading edge marker in the tips of pseudopods. As shown in Figure 2—figure supplement 2H and I, ARL4C knockout did not affect ARPC2 staining and reduced the staining of cortactin and F-actin only modestly. Although F-actin can be utilized to show the leading edge of cells, ARPC2 is also a good hallmark of the leading edge. This may be due to loss of IQGAP1 accumulation by ARL4C depletion, because IQGAP1 is functionally associated with actin assembly. Therefore, loss of ARL4C fairly affects the presence of invasive pseudopods nor the properties that they are the leading edge. Taken together, ARL4C may be necessary for functions of invasive pseudopods rather than their formation. It is quite likely that ARL4C contributes to invasive nature through other than pseudopod formation, and it is reasonable to compare the localization of protein of interest in the same pseudopodial structure between control and ARL4C KO cells. Considering other data in this study, major function of ARL4C in invasive pseudopod would be to recruit MMP14 by binding to IQGAP1. These are described in the text (page 7, lines 16 through 24, page 17, lines 2 through 6)

Reviewer #2 also pointed out the quantification method which is based on “cells presenting protein of interest enriched protrusions.” We have checked whether various treatments given to the cells alter the number of pseudopods per cell as the reviewer mentioned. We could not observe remarkable differences by the treatments (Figure 2—figure supplement 2C and D; Figure 2—figure supplement 2F and G; Figure 4—figure supplement 1B and C; Figure 4—figure supplement 1F and G). Reviewer #2 kindly indicated the possibility that LY294002 could affect formation of pseudopods. However, LY294002 as well as rapamycin was used for inducing PIP3 and PIP2 depletion and only incubated with cells for 30 min followed by fixation. For this reason, there were little effects on the structural formation of pseudopods. The details are described in the last part.

We have also measured the intensity of protein accumulated at the tips of invasive pseudopods, focusing only on cells which form pseudopod(s) (Figure 4—figure supplement 1D). The result was the same as the data using the previous method. However, the presence of pseudopods is dynamic as shown in Figure 2H and Figure 2-video 1 and 2, and the intensity of the tips of pseudopods in a static image does not always reflect the degree of function of protein of interest (POI). For example, low intensity of POI in the tips could indicate not only that it is rarely involved in the function of pseudopods, but also that pseudopod itself is in the retracting phase. Therefore, we decided to use the method in the original manuscript for quantification because it can show intuitively whether POI is localized to the tips and evaluate the whole picture of invasive pseudopods. We think the original method can reflect the invasive ability of the cells more precisely and decided to retain the data. As pointed out by Reviewer #2, this quantification is only reasonable on the assumption that various treatments do not alter the formation of the pseudopods and this was confirmed experimentally as we mentioned above.

We stated that overexpression of ARL4C in BxPC-3 cells induces the membrane protrusions in Figure 2—figure supplement 2E and F and Figure 3M in the first revised manuscript. In addition, since BxPC-3 cells barely express ARL4C, it may be difficult to conclude the action of ARL4C by its overexpression in the context of ARL4C-null cells. I am afraid that the results lead to misunderstanding for our conclusion and would like to study extensively the membrane protrusions induced by ARL4C in BxPC-3 cells in the future. Thus, we would like to remove the data using BxPC-3 cell (Figure 2—figure supplement 2B-F; Figure 3M; Figure 3—figure supplement 1D) of the first revised manuscript. We assume that the changes do not influence our conclusions and hope that the editor and Reviewer #2 could understand our explanation.

For example, PI3K is well known to be required for actin-dependent pseudopod protrusion – so the presentation that LY294002 prevents ARL4C and IQGAP1 accumulation at "protrusions" is not surprising. However, it raises serious questions as to what exactly are the protrusions that are being measured.Without defining exactly what these membrane protrusions and a clear demonstration that the focal adhesion-free protrusions on glass are analagous to protrusive invadopodia, the idea that "Recruitment of KRAS downstream target ARL4C to membrane protrusions accelerates pancreatic cancer cell invasion" is interesting but not supported by the data presented and does not provide a clear mechanistic understanding of the role of ARL4C in cancer invasion. At best there is a correlation of association of ARL4C and as yet to be defined membrane protrusive structures.Here is a list of protrusion data in the paper. Each of these figures should define not only the presence of the protein of interest in protrusions but also the actin labeling and number of protrusions per cell. If these protrusions are defined by the absence of focal adhesions then focal adhesion protein expression in these structures should also be shown. Alternatively, the focus in the manuscript on ARL4C regulation of these poorly defined membrane protrusions should be reduced and emphasis placed on regulation of cancer cell migration, invasion and invadopodia formation (on collagen) in addition to the in vivo data.

The Reviewer #2 commented that figures should define not only the presence of the protein of interest in protrusions but also the actin labeling and number of protrusions per cell. As described above, we showed that the membrane protrusions we defined in the original manuscript are invasive pseudopods containing focal adhesion sites and invadopodia markers or leading edge markers. In addition, we repeated the experiments with overexpression of ARL4C mutants and LY294002 experiments as follows:

When wild-type ARL4C-GFP and ARL4C^Q27L^-GFP were expressed in S2-CP8 cells, they were localized to the tips of pseudopods where cortactin and actin were present. However, ARL4C^T27N^-GFP was not detected in pseudopods although cortactin was observed in the remaining pseudopods. These results suggest that ARL4C is present in the tips of invasive pseudopods where it is expressed as wild type. The results are shown in Author response image 2.

**Author response image 2. sa2fig2:** Effect of ARL4C mutants on pseudopod formation*.* A,B,C, S2-CP8 cells were transfected with the indicated mutants of ARL4C-GFP and stained with the indicated antibodies (A).The percentages of cells with invasive pseudopods compared with the total number of cells were calculated (B). Cells were classified according to the number of pseudopods as indicated (C). A, The regions in the yellow dashed squares are shown enlarged in the left bottom images. The right bottom images are shown in a false color representation of fluorescence intensity. False color representations were color-coded on the spectrum. (B), Data are shown as the mean ± s.d. of 3 biological replicates. *P* values were calculated using one-way ANOVA followed by Bonferroni post hoc test. Scale bars in A, 10 μm. RFI, relative fluorescence intensity. n.s., not significant.

PIP3 is absolutely important for the determination of front-rear polarization of cells and LY294002 treatment would decrease pseudopod formation as the Reviewer #2 commented. In PIP depletion assay shown in Figure 5B, S2-CP8 cells were treated with only 30 min to examine the localization of ARL4C and IQGAP1. Under the conditions, cortactin, a marker of front of the cell, disappeared from the tips of pseudopods along with PIP3 depletion. Since cortactin associates with PIP3, this is not surprising, and serves as a positive control of this assay. Although cortactin lost its localization, protrusive structures still remained 30 min after treatment and it is enough to evaluate the presence of protein of interest. Therefore, we would like to conclude that ARL4C and IQGAP1 disappear from the tips of pseudopods when PIP3 is depleted. When cells were treated with LY294002 for 24 h to test invasive ability, they lost invasive pseudopods and invasive ability. The results are shown in Author response image 3 and described in the text (page 12, lines 17 through 22). We apologize for our insufficient explanation.

**Author response image 3. sa2fig3:** Effect of LY294002 treatment on pseudopod formation*.* A,B, S2-CP8 cells were treated with or without 50 µM LY294002 for 30 min or 24 h before fixation, and stained with the indicated antibodies (A).The percentages of cells with protrusive structures compared with the total number of cells were calculated (B). A, The regions in the yellow dashed squares are shown enlarged in the left bottom images. The right bottom images are shown in a false color representation of fluorescence intensity. False color representations were color-coded on the spectrum. (B), Data are shown as the mean ± s.d. of 3 biological replicates. *P* values were calculated using a two-tailed Student’s t-test. Scale bars in A, 10 μm. RFI, relative fluorescence intensity. n.s., not significant. **, *P* < 0.01.

Again, we really thank the reviewers for providing comprehensive and insightful comments that have helped strengthen our conclusions.